# Learning to Price Homogeneous Data

**Keran Chen**
UW-Madison
kchen429@wisc.edu

**Joon Suk Huh**
UW-Madison
jhuh23@wisc.edu

**Kirthevasan Kandasamy**
UW-Madison
kandasamy@cs.wisc.edu

## Abstract

We study a data pricing problem, where a seller has access to $N$ homogeneous data points (e.g. drawn i.i.d. from some distribution). There are $m$ types of buyers in the market, where buyers of the same type $i$ have the same valuation curve $v_i : [N] \to [0, 1]$, where $v_i(n)$ is the value for having $n$ data points. *A priori*, the seller is unaware of the distribution of buyers, but can repeat the market for $T$ rounds so as to learn the revenue-optimal pricing curve $p : [N] \to [0, 1]$. To solve this online learning problem, we first develop novel discretization schemes to approximate any pricing curve. When compared to prior work, the size of our discretization schemes scales gracefully with the approximation parameter, which translates to better regret in online learning. Under assumptions like smoothness and diminishing returns which are satisfied by data, the discretization size can be reduced further. We then turn to the online learning problem, both in the stochastic and adversarial settings. On each round, the seller chooses an *anonymous* pricing curve $p_t$. A new buyer appears and may choose to purchase some amount of data. She then reveals her type *only if* she makes a purchase. Our online algorithms build on classical algorithms such as UCB and FTPL, but require novel ideas to account for the asymmetric nature of this feedback and to deal with the vastness of the space of pricing curves. Our algorithms achieve $\widetilde{\mathcal{O}}(m\sqrt{T})$ regret in the stochastic setting and $\widetilde{\mathcal{O}}(m^{3/2}\sqrt{T})$ regret in the adversarial setting.

## 1 Introduction

Due to the rise in popularity of machine learning, there is an increased demand for data. However, not all users of data have the wherewithal to collect data on their own, and have to rely on data marketplaces to acquire the data they need. For example, a materials data platform (e.g. [18]), may have collected vast amounts of data from various proprietary sources. Materials scientists in smaller organizations and academia, who do not have large experimental apparatuses, may wish to purchase this data to aid in their research. Similarly, small businesses may wish to purchase customer data for advertising and product recommendations [4, 5], while small technology companies may wish to purchase data about cloud operations to optimize their computing infrastructure [2, 3].

**Model.** Motivated by the emergence of such data marketplaces, we study the following online data pricing problem. A seller has access to $N$ homogeneous data points, (e.g. drawn i.i.d. from some distribution). He wishes to sell the data to a sequence of distinct buyers over $T$ rounds, and intends to achieve large revenue. There are $m$ types of buyers in the data marketplace, with all buyers in type $i$ having the same valuation curve $v_i : [N] \to [0, 1]$ for the data, where $v_i(n)$ represents the buyer's value for having $n$ points. As data is homogeneous, we can treat an agent's value as a function of the *amount of data* $n$ (we will illustrate this in the sequel). Valuation curves are monotone non-decreasing, as more data is better. At each round $t$, the seller chooses a price curve $p_t : [N] \to [0, 1]$, where $p_t(n)$ is the price for purchasing $n$ data points. Then, a buyer with type $i_t$ arrives and purchases an amount of data that maximizes her utility (value minus price), provided that she can achieve non-negative utility. A buyer will reveal her type to the seller *only if* she makes a purchase, and *only after* she

38th Conference on Neural Information Processing Systems (NeurIPS 2024).

makes the purchase. The seller has knowledge of valuation curves of the $m$ types, but does not know the distribution $q$ over types (stochastic setting), or the buyer sequence (adversarial setting). Moreover, he cannot practice non-anonymous (discriminatory) pricing, as he needs to choose the pricing curve $p_t$ without knowledge of the buyer's type on that round.

While there is extensive research on *revenue-optimal pricing* and *learning to price*, data marketplaces merit special attention, both due to their recent emergence and the unique characteristics of data. Typically the number of data $N$ (number of goods) is very large, but data usually satisfies additional properties such as smoothness (an agent's value does not increase significantly with a small amount of additional data) and diminishing returns (additional data is more valuable when a buyer has less data). To illustrate further, note that two steps are essential to develop an effective online learning solution for data pricing. *(1)* First, we need to solve the *planning problem*, i.e. find a revenue-optimal pricing curve when the type distribution $q$ is known. *(2)* Second, when $q$ is unknown, we need to combine the algorithm in step (1) with estimates for $q$ to maximize long-term revenue.

Methods in the existing literature fall short in both steps. *(1)* When the type distribution $q$ is known, the data pricing problem resembles an *ordered item pricing* problem, which is known to be NP-hard [13, 25]. Hence, prior work has aimed at approximating the optimal pricing curves via discretization schemes. Unfortunately, existing discretization schemes have poor, often exponential, dependence on the approximation parameter $\epsilon$. However, achieving sublinear regret in online learning requires choosing $\epsilon$ that vanishes with longer time horizons, i.e. $\epsilon \to 0$ as $T \to \infty$. Therefore, directly using existing discretization schemes in an online setting leads to poor statistical *and* computational properties of the associated online algorithm. This requires us to leverage the above properties of data to design discretization schemes with better dependence on $\epsilon$. *(2)* While there is prior work on learning optimal prices [22, 27, 33], these techniques either fall short of addressing the complexities in our setting, or fail to account for the properties of data, and hence do not scale gracefully when the amount of data $N$ is very large. Moreover, in our online learning setup, the seller faces a trade-off between setting high prices to maximize instantaneous revenue versus setting low prices so as to guarantee a purchase, which results in the buyer revealing their type, which in turn can be helpful in future rounds. Prior work has studied this asymmetric feedback model *only in single-item markets* which is significantly simpler, and *only in the stochastic setting* [23, 47].

## 1.1 Summary of our contributions

Our contributions in this work are threefold: *(1)* First, in §3, we develop discretization schemes for revenue-optimal data pricing under a variety of assumptions, which we will use later in our online learning schemes. *(2)* In §4, we study learning a revenue-optimal price in a stochastic setting, where the customer types on each round are drawn from a fixed but unknown distribution $q$. *(3)* Finally, in §5, we study online learning when the buyer types are chosen by an oblivious adversary.

**1. Discretization (approximation) schemes for revenue-optimal data pricing.** Assuming only monotonicity, we show that there is a discretization of size $\widetilde{\mathcal{O}}((N/\epsilon)^m)$ which is an $\mathcal{O}(\epsilon)$ additive approximation to any pricing curve. When compared to prior work [14, 25], our discretization scheme has smaller dependence on $\epsilon^{-1}$ when the number of types $m$ is small (see Table 1). This will be useful, both statistically and computationally, when we study the online setting, as we need to choose $\epsilon \to 0$ as $T \to \infty$ to achieve sublinear regret. This is still quite large in real-world data marketplaces, where $N$ may be very large. Hence, we also study two other assumptions. First, when valuations are smooth, satisfying an $L$-Lipschitz-like condition, we construct a discretization of size $\widetilde{\mathcal{O}}\left((L/\epsilon^2)^m\right)$, which has no dependence on $N$. Next, under a diminishing returns condition, we construct a discretization of size $\mathcal{O}\left(J^m \epsilon^{-3m} \log^m(N)\right)$, which only has polylogarithmic dependence on $N$.

*Key insights.* We first show that when there are only $m$ types, for any price function $p : [N] \to [0, 1]$, there exists an "$m$–step" price function $p'$ whose revenue is at least as much as that of $p$ on any type distribution $q$. An $m$–step function is non-decreasing and changes values at most $m$ times, allowing us to focus on this restricted class and thereby reduce the search space when $m \ll N$. We then consider discretizations of the data space $[N]$ and valuations $[0, 1]$ which allow us to obtain an $\mathcal{O}(\epsilon)$–approximation to any pricing curve, and then apply this insight to construct our discretizations. Finally, we show that with monotonicity and diminishing returns, similarly accurate approximations are attainable with substantially coarser discretizations.

**2. Learning to price in the stochastic setting.** Next, we turn to the online learning problem described in the beginning in a stochastic setting. On each round, our algorithm computes an

| Algorithm | Assumptions | Size of discretization | Reference |
|---|---|---|---|
| Hartline and Koltun [25] | – | $\widetilde{\mathcal{O}}(2^N \epsilon^{-N})$ | – |
| Chawla et al. [14] | **M** | $N^{\mathcal{O}\left(\epsilon^{-2}\log\epsilon^{-1}\right)}$ | – |
| Algorithm 1 (ours) | **M**, **F** | $\widetilde{\mathcal{O}}(N^m \epsilon^{-m})$ | Theorem 3.1 |
| Algorithm 5 (ours) | **M**, **F**, **S** | $\widetilde{\mathcal{O}}\left(L^m \epsilon^{-2m}\right)$ | Theorem 3.2 |
| Algorithm 2 (ours) | **M**, **F**, **D** | $\widetilde{\mathcal{O}}\left(J^m \epsilon^{-3m}\log^m N\right)$ | Theorem 3.3 |

Table 1: Comparison of discretization (approximation) schemes of prior work and our methods under various assumptions. All methods achieve a $\mathcal{O}(\epsilon)$ additive approximation to any pricing curve. Here, **M** means **M**onotonicity, **F** means that there are a **F**inite ($m$) number of types, **S** means that the valuation curves satisfy a $L$-Lipschitz-like **S**moothness condition (Assumption 1), and **D** means that they satisfy a **D**iminishing returns condition (Assumption 2). The $\widetilde{\mathcal{O}}$ notation suppresses log dependencies when there is already a polynomial dependence on a parameter. Prior work has exponential dependence in either $N$ or $\epsilon^{-1}$. We wish to do better since *(i)* typically, the number of data $N$ is very large and *(ii)* we need $\epsilon \to 0$ as $T \to \infty$ to achieve sublinear regret.

upper confidence bound (UCB) [8, 38] on the revenue for each price curve in the discretization previously developed; we then choose the price curve with the highest UCB. As summarized in Table 2, this algorithm achieves a $\widetilde{\mathcal{O}}(m\sqrt{T})$ bound on the regret for *any* discretization scheme, including those from prior work. In the stochastic setting, the key advantage of our discretization schemes is computational.

*Key insights.* Both the design and the anlaysis of an algorithm is challenging in this setting due to two reasons: *(i)* the large size of the discretization and *(i)* the asymmetric nature of feedback. First, naively maintaining UCBs for each price leads to large confidence intervals, and hence large regret as the size of the discretization is large; instead, we construct confidence intervals on estimates of the type distribution, and translate them to UCBs for the revenue. Second, the asymmetric nature of the feedback places us between bandit and full-information settings. Treating this like a bandit setting would lead to poor, exponential dependence on $m$ in the regret. However, we are unable to treat this as full information since the type distribution is revealed only if there is a purchase. Handling this asymmetry requires a delicate construction of the UCB.

**3. Learning to price in the adversarial setting.** We study learning in an adversarial setting where the types on each round may be chosen adversarially. Table 2 shows the regret and time complexity of our method when paired with various discretization schemes. In the adversarial setting, our discretization schemes offer both computational and statistical advantages compared to prior work.

*Key insights.* Our algorithm builds on the Follow-the-Perturbed-Leader (FTPL) [31], originally designed for full-information settings and not directly applicable here. To handle asymmetric feedback, we use the information we have about the valuation curves to keep track of which customers would not have made a purchase given a price curve. If a purchase is made and we observe feedback, we use the usual FTPL update, but if not, we reward each pricing curve with the sum of revenue of all types that would not purchase in that current round.

## 1.2 Related work

**Dynamic pricing**. The online posted-price mechanism, also known as dynamic pricing, is a central research area in algorithmic market design [19, 33]. In the most classical setting [33], the seller sets a price for an item in each round, and a buyer purchases the item only if their valuation exceeds the posted price. While several extensions of this setting have been explored for both parametric [12, 20, 28, 29, 32, 46] and non-parametric [11, 17, 39, 40, 44] demands, most focus on single-parameter demands, i.e., selling a single item to buyers. Our data pricing problem is multi-parameter, as demands are parameterized by multiple outcomes, i.e. the number of data points.

**Bayesian unit-demand pricing problem**. Formally, our data pricing problem is a variant of the Bayesian Unit-demand Pricing Problem (BUPP) [13]. BUPP addresses the problem of (offline) revenue maximization over a known distribution of unit-demand buyers, meaning they want to buy at

| Setting | Assumptions | Regret bound | Complexity per iteration | Reference |
|---|---|---|---|---|
| Stochastic | **M, F** | $\widetilde{O}\left(m\sqrt{T}\right)$ | $\widetilde{O}\left(\left(\frac{N}{m}\right)^m T^{m/2}\right)$ | Theorem 4.1 |
| | **M, F, S** | | $\widetilde{O}\left((LT)^m\right)$ | |
| | **M, F, D** | | $\widetilde{O}(J^m T^{3m/2})$ | |
| Adversarial | **M, F** | $\widetilde{O}\left(m^{3/2}\sqrt{T}\right)$ | $\widetilde{O}\left(\left(\frac{N}{m}\right)^m T^{m/2}\right)$ | Theorem 5.1 |
| | **M, F, S** | | $\widetilde{O}\left((LT)^m\right)$ | |
| | **M, F, D** | | $\widetilde{O}\left(J^m T^{3m/2}\right)$ | |

| Discretization method | Assumptions | Complexity per iteration | Regret (Adversarial) |
|---|---|---|---|
| Hartline and Koltun [25] | **F** | $\widetilde{O}(2^N \epsilon^{-N})$ | $\widetilde{O}(m\sqrt{TN})$ |
| Chawla et al. [14] | **M, F** | $N^{O\left(\epsilon^{-2}\log\epsilon^{-1}\right)}$ | $\widetilde{O}\left(mT^{3/4}\right)$ |

Table 2: Comparison of regret and time complexity of *our* online learning methods when paired with our discretization schemes and schemes from prior work. See Table 1 for a description of the assumptions. All methods, including [14, 25] achieve $O(m\sqrt{T})$ regret in the stochastic setting.

most one item from the inventory. In BUPP, a seller has $N$ distinct items to sell to a unit-demand buyer whose valuations are $v = (v_1, \ldots, v_N)$, where $v_i$ is the value of the $i$th item. Given prices $p_i$, $i \in [N]$, the unit-demand buyer purchases a single item $i \in [N]$ that maximizes their utility: $v_i - p_i$. Assuming the valuation profile $v$ follows a known distribution $D$, the goal of BUPP is to find the best prices $\{p_i\}_{i \in [N]}$ that maximize the seller's expected revenue.

Our data pricing problem is a variant of BUPP in two ways: *(1)* We study the sequential setting where type distributions are *unknown*, while valuation profiles for each type are known, and *(2)* We assume monotonic values $v_1 \leq \cdots \leq v_N$, which is natural in data pricing. Unfortunately, BUPP is a computationally intractable problem, as is ours. BUPP is known to be NP-hard even when $D$ is a product distribution [16]. Moreover, even assuming that values are monotonic (i.e., $v_1 \leq \cdots \leq v_N$), the problem remains (strongly) NP-hard [14]. Therefore, we aim to provide a reasonably efficient no-regret algorithm for our problem, especially when the number of types $m$ is a fixed constant.

The previous works most relevant to our paper are Hartline and Koltun [25] and Chawla et al. [14], which study offline revenue maximization for unit-demand buyers. Buyers in our problem are also unit-demand, as each amount of data points can be seen as an individual item. Revenue maximization for unit-demand buyers is known to be computationally intractable [24], even with ordered (monotonic) buyer values [14], leading these works to focus on approximation algorithms. Hartline and Koltun [25] proposed an approximation algorithm with near-linear runtime in the number of buyers, given a fixed number of items. Chawla et al. [14] introduced a polynomial-time approximation scheme (PTAS) for unit-demand buyers with monotonic values. In this work, we extend the framework to the online setting with partial feedback, which has more practical implications.

In addition, Balcan and Beyhaghi [10] provide new guarantees for learning revenue-maximizing menus of lotteries and two-part tariffs, demonstrating that their discretization technique yields efficient solutions for specific pricing models. Similar discretization methods could be investigated in future work to potentially improve our approach in more complex data pricing scenarios.

**Market design for data-sharing**. In recent years, there has been a plethora of work devoted to algorithmic market design for data sharing [6, 7, 30, 43]. These works provide ingenious solutions to challenges unique to the data market, such as free replicability and the difficulty of valuation due to the combinatorial nature of data. Except for Agarwal et al. [6], the above-cited solutions are inherently *offline or single-shot*. While we focus on a simplified yet relevant setting where data comes from a single source, resulting in monotonic valuations, in this work, we tackle the problem in a sequential, dynamic setting, which has practical importance. In contrast to our approach, Agarwal et al. [6] considered the price to be a constant (i.e., a scalar rather than a price vector) to address the inherent computational intractability of multi-dimensional pricing. Instead, we maintain the price

as a vector (i.e., a price function) but focus on cases where the valuation function satisfies natural properties such as monotonicity, smoothness, and diminishing returns.

## 2  Problem setting, assumptions, and challenges

A seller has $N$ homogeneous data points. There are $m$ types of buyers who wish to purchase this data. A buyer of type $i \in [m]$ has a valuation curve $v_i : [N] \to [0, 1]$, where $v_i(n)$ is her value for $n$ data points. We will assume $v_i(n)$ is non-decreasing as more data is valuable, and further that $v_i(0) = 0$.

**Example 1.**  To motivate this model, consider a seller with $N$ ordered data points $\{x_1, \ldots, x_N\}$, drawn i.i.d. from a distribution $D$. If a buyer purchases $n$ points, she receives the first $n$ points, $X_n = \{x_1, \ldots, x_n\}$. Her *ex-post* value $\widetilde{v}_i(X_n)$ may represent the accuracy of her ML model trained with $X_n$. However, as the buyer has not seen the data before the purchase, she does not know which specific points she will receive, and hence her (*ex-ante*) value $v_i(n) = \mathbb{E}_{X_n}[\widetilde{v}_i(X_n)]$ is the expected model accuracy when $n$ i.i.d points are drawn from $D$. The different types could be buyers who use the data for different tasks or models. For instance, with ImageNet's [21], $N \approx 1.4$ million data points, different types of buyers could perform different learning tasks such as object detection, identification, and segmentation, and/or train different models such as AlexNet [36], ResNet [26], and GoogLeNet [42]. Both empirically and theoretically, for many learning tasks, $v_i(n)$ is non-decreasing, and satisfies additional characteristics such as smoothness and/or diminishing returns.

**Pricing curves, buyer utility, and buyer purchase model.**  Let $p : [N] \to [0, 1]$ be a pricing curve chosen by the seller. Let $\mathcal{P} \triangleq \{p : [N] \to [0, 1] : p(0) = 0\}$ denote the set of all pricing curves. If a buyer purchases $n$ points, her utility is $u_i(n) = v_i(n) - p(n)$. If a buyer can achieve non-negative utility, i.e. $v_i(n) \geq p(n)$ for some $n \in [N]$, she will purchase an amount of data to maximize her utility. To fully specify the buyer's purchase model, we will assume that when there are multiple $n$ which maximizes her utility, she will choose the largest such $n$. Formally, for a given pricing curve $p$, a buyer of type $i$ will purchase $n_{i,p}$ points where,

$$n_{i,p} \triangleq \begin{cases} 0 & \text{if } v_i(n) < p(n) \text{ for all } n \in [N], \\ \max\left\{\operatorname{argmax}_{n \in [N]}\left(v_i(n) - p(n)\right)\right\} & \text{otherwise.} \end{cases} \tag{1}$$

*Optimal revenue.*  It follows that the revenue from a buyer of type is $p(n_{i,p})$. Let $q = (q_1, \ldots, q_m)$ be the distribution of the buyers. Under this distribution $q$, the expected revenue $\operatorname{rev}(p)$ for a price curve $p$, the optimal price $p^{\text{OPT}}$, and the optimal revenue OPT as follows:

$$\operatorname{rev}(p) \triangleq \sum_{i=1}^{m} q_i \cdot p(n_{i,p}), \qquad p^{\text{OPT}} \triangleq \operatorname*{argmax}_{p \in \mathcal{P}} \operatorname{rev}(p), \qquad \text{OPT} \triangleq \operatorname{rev}(p^{\text{OPT}}). \tag{2}$$

We have omitted the dependence on $q$ in $\operatorname{rev}$, $p^{\text{OPT}}$, and OPT. There is no closed-form solution to finding the optimal pricing curve, even when $q$ is known. Therefore, in §3, we explore discretization methods to approximate $p^{\text{OPT}}$, which will then be used in §4 and §5 to develop online learning algorithms. Unfortunately, the size of this discretization can be very large in $N$ and $m$ without further assumptions. Therefore, we also consider two additional commonly satisfied conditions by data.

Our first such assumption states that buyer valuation curves satisfy a Lipschitz-like smoothness condition with Lipschitz constant $L/N$. We use $L/N$ instead of $L$ since the number of data has a range $[0, N]$, while the valuations only have a range $[0, 1]$. This condition states that a buyer's valuation does not change significantly if she only purchases a few additional points.

**Assumption 1** (Smoothness, **S**).  *For all $n, n' \in [N]$, we have $v_i(n + n') - v_i(n) \leq \frac{L}{N} n'$.*

Our second condition is based on the fact that data typically exhibits diminishing returns [34, 35]. This means that an additional data point is more valuable when there is less data, i.e. $v_i(n+1) - v_i(n)$ is decreasing with $n$. We will in fact make a stronger assumption, and justify it below.

**Assumption 2** (Diminishing returns, **D**).  *There exists some $J > 0$ such that, for all types $i \in [m]$, and for all $n \in [N]$, we have $v_i(n + 1) - v_i(n) \leq \frac{J}{n}$.*

Assumption 2 quantifies the rate of decrease of diminishing returns. Following Example 1, the valuation (accuracy) curves for many learning problems take the form $v_i(n) = \alpha - \beta n^{-\gamma}$; for

instance, for binary classification in a VC class $\mathcal{H}$, $\alpha$ may be the best accuracy in $\mathcal{H}$, $\beta \in \mathcal{O}(\sqrt{d_{\mathcal{H}}})$ where $d_{\mathcal{H}}$ is the VC dimension, and $\gamma = 1/2$ [41]; similarly, for nonparametric regression of a twice differentiable function, $\alpha$ and $\beta$ are constants while $\gamma = 2/5$ [45]. In such cases, Assumption 2 is satisfied with $J = \beta\gamma$. Note that neither assumption subsumes the other: a non-concave Lipschitz function will not satisfy Assumption 2, while a suitable $L$ for a function which satisfies Assumption 2 may need to be very large for Assumption 1 to hold for small $n$.

## 2.1 Learning to price in online settings

In this work, we will also study how a seller may learn to maximize revenue. In our learning problem, the seller is aware of the valuation curves $\{v_i\}_i$ of each type, but does not know the distribution of types (stochastic setting) or there may be no such distribution (adversarial setting).

**Setup.** The seller repeats the data market for $T$ rounds. At the beginning of each round, he chooses some price curve $p_t \in \mathcal{P}$. *After* the seller has chosen $p_t$, a new buyer of type $i_t \in [m]$ appears and purchases $n_t = n_{i_t, p_t}$ amount of data (see (1)). The buyer is aware of her own valuation curve. If she makes a purchase, that is if $n_t > 0$, she pays $p_t(n_t)$ to the seller and reveals her type $i_t$. Otherwise, the buyer will make no payment and not reveal her type.

We have assumed that *a priori*, the seller is aware of the buyer valuation curves $\{v_i\}_{i \in [m]}$, and that buyers are aware of their own valuation curves. In Example 1, a seller can profile how different machine learning models perform with different amounts of data and publish them ahead of time. The buyers can also gauge their value from these curves, even though they do not have access to the data. Next, we have also assumed that buyers will reveal their type after the purchase. In modern machine learning as a service platforms [1, 4, 18], buyers directly run their jobs in the seller's computing platform, so the seller can observe the buyers job *type* directly. Even if this is not the case, sellers can elicit this information via questionnaires and reviews from customers who have made a purchase [23].

**Challenges.** Despite these assumptions, the learning problem remains challenging for two main reasons. First, the space of price curves is vast: discretizing the valuations in $[0, 1]$ into $K$ bins, still leaves $\mathcal{O}(K^N)$ possible price curves, which is both statistically and computationally intractable, especially for large $N$. Second, in addition to the exploration-exploitation trade-off usually encountered in sequential decision-making, the seller faces a tension between high instantaneous revenue and information acquisition: setting high prices can yield high immediate revenue if a purchase occurs, but it also increases the risk of no purchase, resulting in no revenue and crucially no feedback about the buyer type which could help him in future rounds. This trade-off was recently studied for single-item markets in a stochastic setting [23, 47], but is more complex in our multi-item problem. Moreover, to our knowledge, no existing work addresses this asymmetric feedback model in an adversarial setting, even for single-item markets. Next, we describe the buyer arrival model and define the regret for the learning problem in both stochastic and adversarial settings.

**Stochastic setting.** Here, there is some fixed but unknown distribution of types $q$. On each round, a buyer of type $i_t \sim q$ is drawn independently. The optimal expected revenue OPT under type distribution $q$ is as defined in (2). The regret $R_T$ is as defined below. We wish to design algorithms which have small expected regret $\mathbb{E}[R_T]$, where the expectation accounts for both the sampling of types $i_t \sim q$ and any randomness in the algorithm. We have,

$$R_T \triangleq T \cdot \text{OPT} - \sum_{t=1}^{T} p_t(n_t) = T \cdot \text{OPT} - \sum_{t=1}^{T} p_t(n_{i_t, p_t}). \tag{3}$$

**Adversarial setting.** Here, the types on each round $\{i_t\}_{t=1}^{T}$ are chosen arbitrarily, possibly by an oblivious adversary, ahead of time. The type on round $t$ is revealed to the seller only at the end of the round, and only if there is a purchase. In the adversarial setting, we define our regret $R_T$ with respect to the single best price in $\mathcal{P}$ in hindsight. We wish to design algorithms with small expected regret $\mathbb{E}[R_T]$, where the expectation is with respect to any randomness in the algorithm. We have,

$$R_T \triangleq \max_{p \in \mathcal{P}} \sum_{t=1}^{T} p(n_{i_t, p}) - \sum_{t=1}^{T} p_t(n_{i_t, p_t}). \tag{4}$$

---

**Algorithm 1** Price discretization scheme under monotonicity

---

**Given:** Approximation parameter $\epsilon > 0$.
Let $W$ be discretization of the valuation space $[0, 1]$ defined as follows,

$$Z_i \triangleq \left\{ \epsilon(1 + \epsilon)^i; \quad \forall\, i \in \left\{ 0, 1, \ldots, \left\lceil \log_{1+\epsilon} \frac{1}{\epsilon} \right\rceil \right\} \right\},$$

$$W_i \triangleq \left\{ Z_{i-1} + Z_{i-1} \cdot \frac{\epsilon k}{m}; \quad \forall\, k \in \{1, 2, ..., \lceil (2 + \epsilon)m \rceil\} \right\}, \quad W \triangleq \bigcup_{i=1}^{\left\lceil \log_{1+\epsilon} \frac{1}{\epsilon} \right\rceil} W_i.$$

Set $\overline{\mathcal{P}}$ to be the class of all "$m$-step" functions mapping $[N]$ to $W$.

---

## 3 Efficient discretization of price curves with small errors

We first study the revenue maximization problem in the offline setting, where the seller knows both the valuation curves $v_i, i \in [m]$, and the type distribution $q$. Our goal is to design a discretization so as to achieve revenue within a gap of $\mathcal{O}(\epsilon)$ from OPT. Before discussing our discretization algorithms, we first show that the optimal pricing curve is "simple" when there are at most $m$ types.

**Lemma 3.1.** *Assume there are $m$ types with non-decreasing value curves $\{v_i\}_{i \in [m]}$. For any non-decreasing price curve $p$, there exists an "$m$-step" price curve $\bar{p}$ that yields expected revenue at least that of $p$ with respect to any distribution over the $m$ types. Here, $m$-step refers to non-decreasing functions $f : [N] \to [0, 1]$ where $f(n + 1) - f(n) > 0$ in at most $m$ points (i.e., at most $m$ jumps).*

Lemma 3.1, proven in Appendix A.1, will be an important tool in all three discretization algorithms of this section. It will allow us to reduce the space of pricing curves as we only need to focus on $m$-step price curves. Next, we present our first discretization procedure in Algorithm 1, which only assumes the monotonicity of the valuation curves.

**Discretization scheme under monotonic valuations.** Our discretization proecdure, outlined in Algorithm 1, adapts the method in Hartline and Koltun [25] using Lemma 3.1. For this, we will first construct a discretization $W$ of the valuation space as follows. Let $Z_i = \epsilon(1 + \epsilon)^i$, $i = 0, 1, \ldots, \left\lceil \log_{1+\epsilon} \frac{1}{\epsilon} \right\rceil$ be the powers of $(1 + \epsilon)$ on price space $[\epsilon, 1]$. For each $i$, we let $W_i$ be a uniform discretization of the interval $[Z_{i-1}, Z_{i+1})$ uniformly with gap $Z_{i-1} \cdot \frac{\epsilon}{m}$. Finally, let $W$ be the union of all such $W_i$. According to Lemma 3.1, every price function in $\mathcal{P}$ has the same revenue as an $m$-step function. We set $\overline{\mathcal{P}}$ to be all choices of non-decreasing $m$-step functions that take value in $W$. We have the following theorem about Algorithm 1 which we prove in Appendix A.2.

**Theorem 3.1.** *Consider the discretization $\overline{\mathcal{P}}$ as constructed in Algorithm 1. For any type distribution, there exists $p \in \overline{\mathcal{P}}$ such that $\mathrm{rev}(p) \geq \mathrm{OPT} - \mathcal{O}(\epsilon)$. Moreover, we have $|\overline{\mathcal{P}}| \leq \left( \frac{e(N-1)}{m} \right)^m \left( e \lceil (2 + \epsilon) \rceil \left\lceil \log_{1+\epsilon} \frac{1}{\epsilon} \right\rceil \right)^m \in \widetilde{\mathcal{O}}\left( \left( \frac{N}{\epsilon} \right)^m \right).$*

**Discretization scheme for smooth monotonic valuations.** Due to space constraints, we present our algorithm, under Assumption 1 in Appendix A.4. We have the following theorem about Algorithm 5.

**Theorem 3.2.** *Consider the discretization $\overline{\mathcal{P}}$ as constructed in Algorithm 5. Under Assumption 1, for any type distribution, there exists $p \in \overline{\mathcal{P}}$ such that $\mathrm{rev}(p) \geq \mathrm{OPT} - \mathcal{O}(\epsilon)$. Moreover, $|\overline{\mathcal{P}}| \in \mathcal{O}\left( \log_{1+\epsilon}^m (1/\epsilon) \cdot (L/\epsilon)^m \right) \in \widetilde{\mathcal{O}}\left( \left( \frac{L}{\epsilon^2} \right)^m \right).$*

**Discretization scheme for monotone valuations under diminishing returns.** Finally, we study discretization schemes under the diminishing returns condition. Our procedure, outlined in Algorithm 2 proceeds as follows. We use the same discretization $W$ of the valuation space from Algorithm 1. Next, we will discretize the dataspace $[N]$. To exploit the structure in the diminishing returns condition, we will need to do so more densely when $n$ is small. For this, let $Y_i = \frac{2Jm}{\epsilon^2}(1 + \epsilon^2)^i$, $i = 0, \ldots, \left\lceil \log_{1+\epsilon^2} \frac{N\epsilon^2}{2Jm} \right\rceil$ be the powers of $(1 + \epsilon^2)$ on data space $\left[ \frac{2Jm}{\epsilon^2}, N \right]$. For each $i$, the set $Q_i$ further partitions the interval $[Y_i, Y_{i+1})$ uniformly with gap $Y_i \cdot \frac{\epsilon^2}{2Jm}$. For $n$ smaller than $\frac{2Jm}{\epsilon^2}$, we do not discretize it as the valuations may change rapidly when $n$ is small. Let $N_{\mathbf{D}}$ be the union of

---
**Algorithm 2** Price discretization scheme monotonic valuations under diminishing returns
---

**Given:** Diminishing returns constant $J$, approximation parameter $\epsilon$.

Let $W \triangleq \bigcup_{i=2}^{\lceil \log_{1+\epsilon} \frac{1}{\epsilon} \rceil} W_i$, were $W_i$s are the same as in Algorithm 1.

Let $N_{\mathbf{D}}$ be discretization of the interval $[0, N]$ defined as follows,

$$Y_i \triangleq \left\lfloor \frac{2Jm}{\epsilon^2}(1+\epsilon^2)^i \right\rfloor, \quad i = 0, 1, \ldots, \left\lceil \log_{1+\epsilon^2} \left( \frac{N\epsilon^2}{2Jm} \right) \right\rceil,$$

$$Q_i \triangleq \left\{ \left\lfloor Y_i + Y_i \cdot \frac{\epsilon^2 k}{2Jm} \right\rfloor, \quad k = 0, 1, \ldots, \lfloor 2Jm \rfloor \right\}, \quad Q \triangleq \bigcup_{i=1}^{\left\lceil \log_{1+\epsilon^2} \left( \frac{N\epsilon^2}{2Jm} \right) \right\rceil} Q_i,$$

$$N_{\mathbf{D}} \triangleq \left\{ 1, 2, \ldots, \left\lfloor \frac{2Jm}{\epsilon^2} \right\rfloor \right\} \cup Q.$$

The discretization price set $\overline{\mathcal{P}}$ is the class of all "$m$-step" price curves on function space $N_{\mathbf{D}} \to W$.

---

$\left\{ 1, 2, \ldots, \left\lfloor \frac{2Jm}{\epsilon^2} \right\rfloor \right\}$ and all the set $Q_i$. Therefore, $N_{\mathbf{D}}$ has a size of at most $\frac{2Jm}{\epsilon^2} + 2Jm \lceil \log_{1+\epsilon^2} \frac{N\epsilon^2}{2Jm} \rceil$. We have the following theorem about Algorithm 2 which we prove in Appendix A.5.

**Theorem 3.3.** *Consider the discretization $\overline{\mathcal{P}}$ as constructed in Algorithm 2. Under Assumption 2, for any type distribution, there exists $p \in \overline{\mathcal{P}}$ such that $\mathrm{rev}(p) \geq \mathrm{OPT} - \mathcal{O}(\epsilon)$. Moreover,*

$$|\overline{\mathcal{P}}| \in \mathcal{O}\left( \left( \frac{J}{\epsilon^2} \right)^m \log^m \left( \frac{N\epsilon^2}{Jm} \right) \cdot \left( \log_{1+\epsilon}^m 1/\epsilon \right) \right) \in \widetilde{\mathcal{O}}\left( \left( \frac{J}{\epsilon^3} \right)^m \right).$$

*Proof outline.* By Lemma 3.1, we may assume the optimal price curve $p^\star = \{(n_i^\star, p_i^\star)\}_{i=1}^m$ is an $m$-step function, where $p_i^\star$ denote the value of $p$ on step $i$. We generate an $m$-step price curve $p = \{(n_i, p_i)\}_{i=1}^m$ on space $N_{\mathbf{D}} \to W$ such that $n_i$ is obtained by rounding down $n_i^\star$ to the closest value in $N_{\mathbf{D}}$, and $p_i \geq p_i^\star/(1+\epsilon)$. We then show that if a buyer purchases at step $i$ under price $p^\star$, she will not purchase at step $j < i$ under new price $p$. Therefore, the revenue from this buyer is at least $p_i \geq p_i^\star/(1+\epsilon) = p_i^\star - \mathcal{O}(\epsilon)$, which ensures that $\mathrm{rev}(p) \geq \mathrm{OPT} - \mathcal{O}(\epsilon)$.

## 4   Online learning in the stochastic setting

We now study the online learning problem outlined in §2.1 in the stochastic setting. Our Algorithm, outlined in Algorithm 3 is based on the classical upper confidence bound (UCB) algorithm for stochastic bandits [8, 38]. It takes a discretization $\overline{\mathcal{P}}$ of the pricing curves as input, and on each round chooses a $p_t \in \overline{\mathcal{P}}$ which has the largest UCB on the revenue.

The key challenge lies in constructing an UCB. As $\overline{\mathcal{P}}$ is large, naively constructing UCB over prices in $\overline{\mathcal{P}}$ will lead to a $\sqrt{|\overline{\mathcal{P}}|T \log T}$ upper bound, leading to poor, exponential dependence on $m$. This is the bound if we only observe the reward for the prices that are actually pulled, but do not observe the types after purchase. Therefore, naively applying UCB is like bandit feedback. On the other extreme, had we been in an alternative setting where we observe the type regardless of purchase, this is like a full information feedback because once observe the type, we know the revenue for all prices. Then UCB gives us $\sqrt{\log(|\overline{\mathcal{P}}|)T \log T}$ upper bound. We are in an intermediate regime between bandit feedback and full information: The challenge in constructing the UCB arises because we only observe types upon purchase. As the key unknown is the type distribution, we maintain UCBs for it and translate them to UCBs for the revenue. In particular, our UCB depends on how many times a buyer *could* have purchased at a given round, which is a random quantity depending on the algorithm itself.

*Construction of UCB.* We will now show how to construct the upper confidence bound $\widehat{\mathrm{rev}}_t$ at the end of round $t$, which will be used in computing $p_{t+1}$. For $\tau \leq t$, let $S_\tau$, defined below in (5), be the set of types who would have purchased in round $\tau$ at price $p_\tau$ had they appeared in that round. Then, for any type $i \in [m]$, we define $T_{i,t}$ to be the number of times that type $i$ appears in set $S_\tau$ for

---
**Algorithm 3** Online data pricing in the stochastic setting.
---
**Given:** time horizon $T$, discretization $\overline{\mathcal{P}}$ of price curves.
Set $p_1$ to be the zero function.       # Give data away for free on round 1.
A buyer of type $i_1 \sim q$ arrives and purchases $N$ data points at price 0.
**for** $t = 2$ to $T$ **do**
     Compute the UCB $\widehat{\text{rev}}_{t-1}(p)$ on the revenue of $p$ for each $p \in \overline{\mathcal{P}}$.      # See (5), (6), and (7).
     Set $p_t = \text{argmax}_{p \in \overline{\mathcal{P}}} \widehat{\text{rev}}_{t-1}(p)$.
     A buyer of type $i_t \sim q$ arrives, purchases $n_{i_t,p_t}$ points, and pays $p_t(n_{i_t,p_t})$.
**end for**
---

$\tau \in \{1, \ldots, t\}$. That is, $T_{i,t}$ measures the number of times a buyer of type $i$ would have purchased during the first $t$ rounds. We have,

$$S_\tau \triangleq \big\{ i \in [m] : \exists n \in [N], v_i(n) - p_\tau(n) \geq 0 \big\}, \qquad T_{i,t} \triangleq \sum_{\tau=1}^{t} \mathbb{I}(i \in S_\tau). \tag{5}$$

Note that as we use the 0 price function on round 1, i.e. $p_1(\cdot) = 0$, we have $T_{i,t} > 0$ for all $t > 1$. Next, we estimate $q_i$ via the fraction of times that type $i$ has appeared in the past $t$ rounds, provided that $i \in S_\tau$ for $\tau \in \{1, \ldots, t\}$. We have defined this quantity, $\overline{q}_{i,t}$ below in (6). Via a standard application of Hoeffding's inequality, we can show that $\big| q_i - \overline{q}_{i,t} \big| \leq \sqrt{(\log T)/T_{i,t}}$ with high probability. Using this, we can construct an upper confidence bound $\widehat{q}_{i,t}$ as follows,

$$\overline{q}_{i,t} \triangleq \frac{1}{T_{i,t}} \sum_{\tau=1}^{t} \mathbb{I}(i \in S_\tau, i_\tau = i), \qquad \widehat{q}_{i,t} \triangleq \overline{q}_{i,t} + \sqrt{\frac{\log T}{T_{i,t}}}. \tag{6}$$

We now translate the UCBs on $q$ to the UCBs on the revenue. Recall from (1) that a buyer of type $i$ will purchase $n_{i,p}$ points at price $p$ and the revenue from this buyer will be $p(n_{i,p})$. Note that as the seller has access to the valuation curves, he can compute $n_{i,p}$ for any $i$ and price curve $p$. Since $\text{rev}(p) = \mathbb{E}_{i \sim q}[p(n_{i,p})]$, we have the following natural UCB for $\text{rev}(p)$ on round $t$:

$$\widehat{\text{rev}}_t(p) \triangleq \sum_{i=1}^{m} \widehat{q}_{i,t} \cdot p(n_{i,p}). \tag{7}$$

This completes the description of our construction. The following theorem bounds the regret for Algorithm 3 when paired with any of the discretization schemes in §3. While the computational complexity of our method depends on $|\mathcal{P}|$, there is no dependence on the regret because of the above construction of the UCB. The proof is given in Appendix C.

**Theorem 4.1.** *Suppose in Algorithm 3 we use a discretization $\overline{\mathcal{P}}$ which is a $\mathcal{O}(1/\sqrt{T})$ additive approximation to any price curve. Then, the regret of Algorithm 3 satisfies $\mathbb{E}[R_T] \in \widetilde{\mathcal{O}}(m\sqrt{T})$.*

*Proof challenges.* When bounding the regret, we first observe that the subsets $S \subset [m]$ induces a partitioning of the price curves, where $p$ belongs to the partition of $S$, if all types in $S$ would make a purchase at price $p$, and all types in $S^c$ would not make a purchase at price $p$. With this insight, we can view the action of a seller as not just choosing a price curve, but also choosing a set $S_t \subset [n]$. That is, $S_t$ can be viewed as a super-arm in a combinatorial semi-bandit problem [37].

## 5 Online learning in the adversarial setting

We now study the adversarial setting. Similar to the stochastic setting, our algorithm will use a discretization of the price curves from §3. We will control regret by bounding both the discretization error and the algorithm's regret relative to the best pricing curve in the discretization.

Before proceeding, let us first contextualize our feedback model against prior work. If the buyers do not reveal their types, this becomes an adversarial bandit problem with $|\overline{\mathcal{P}}|$ arms (pricing curves) [33]. Using an algorithm such as EXP-3 [9] results in large $\widetilde{\mathcal{O}}(T^{1/2}|\overline{\mathcal{P}}|^{1/2})$ regret, which is not ideal due to $|\overline{\mathcal{P}}|$'s exponential dependence in $m$. Conversely, if buyers reveal their types regardless of purchase,

---

**Algorithm 4** Online data pricing in the adversarial setting.

---

**Given:** time horizon $T$, discretization $\overline{\mathcal{P}}$, perturbation parameter $\theta$.
For each $p \in \overline{\mathcal{P}}$, sample $\theta_p$ from an exponential distribution with pdf $\theta e^{-\theta x}$
**for** $t = 1$ to $T$ **do**

   Set price curve for the current round $\quad p_t = \underset{p \in \overline{\mathcal{P}}}{\operatorname{argmax}} \sum_{\tau=1}^{t-1} r_\tau(p) + \theta_p$.
   A buyer of type $i_t$ arrives, purchases $n_{i_t, p_t}$ points, and pays $p_t(n_{i_t, p_t})$.
   **if** $n_{i_t, p_t} > 0$ **then**   Set $r_t(p) = p(n_{i_t, p})$ for all $p \in \overline{\mathcal{P}}$.       # If there was a purchase
   **else**   Set $r_t(p) = \sum_{i \in S_t^c} p(n_{i,p})$ for all $p \in \overline{\mathcal{P}}$.       # See (5) for $S_t$.
   **end if**
**end for**

---

this is equivalent to full information feedback, where algorithms such as Hedge or Follow-the-perturbed-leader (FTPL) [31] yield $\mathcal{O}(T^{1/2} \log^{1/2} |\overline{\mathcal{P}}|)$ regret, translating to $\widetilde{\mathcal{O}}((mT)^{1/2})$ with our discretization schemes in §3. In our intermediate regime, where feedback is only revealed upon purchase, we aim for a middle ground. We show our algorithm, outlined in Algorithm 4, achieves $\widetilde{\mathcal{O}}(m^{3/2}T^{1/2})$ regret, which is worse than full information, but still depends polynomially on $m$.

Our algorithm takes a discretization $\overline{\mathcal{P}}$ and a perturbation parameter $\theta$ as input. First, it samples a random perturbation $\theta_p$ from an exponential distribution with pdf $\theta e^{-\theta x}$ for each pricing curve $p$ in $\overline{\mathcal{P}}$. It maintains rewards $\{r_t(p)\}_{t,p}$ for each round $t$ and price curve $p$. On each round, it chooses the price curve that maximizes the perturbed cumulative reward $\sum_{\tau=1}^{t} r_\tau(p) + \theta_p$.

This scheme is similar to FTPL, but the key difference is in how we design the rewards $\{r_t(p)\}_{t,p}$. To describe this, let $S_t$, defined exactly as in (5), be the set of agents who would have purchased in round $t$ at price $p_t$. At the end of the round, if there was a purchase, for all prices $p \in \overline{\mathcal{P}}$, we set the reward to be $r_t(p) = p(n_{i_t, p})$, i.e. the payment we would have received from the buyer at that round, had the price been $p$ (see (1)). If there was no purchase, we know that $i_t \notin S_t$, in which case we set $r_t(p) = \sum_{i \in S_t^c} p(n_{i,p})$. In this case, $r_t(p)$ is an upper bound on $p(n_{i_t, p})$, and this upper bound is tight around prices similar to the chosen price $p_t$; in fact, $r_t(p_t) = 0$ if there was no purchase. Intuitively, $r_t(p)$ deals with the uncertainty of not knowing the type on round $t$ by providing a large reward (as we are taking the sum) to prices that *could have* resulted in a purchase, which encourages exploration of such prices in future rounds. This intuition will help us bound the regret.

Theorem 5.1 provides a bound on the regret for Algorithm 4. Its proof is given in Appendix B. Combining this with the size of $\overline{\mathcal{P}}$ under the various assumptions in §3, we obtain $\widetilde{\mathcal{O}}(m^{3/2}\sqrt{T})$ regret.

**Theorem 5.1.** *Suppose in Algorithm 4 we use a discretization $\overline{\mathcal{P}}$ which is a $\mathcal{O}(1/\sqrt{T})$ additive approximation to any price curve. Let $R_T$ be as defined in (4). Then, for Algorithm 4, we have $\mathbb{E}[R_T] \in$* $\mathcal{O}\left(m^2 \theta T + \theta^{-1}\left(1 + \log |\overline{\mathcal{P}}|\right)\right)$. *Setting $\theta = \sqrt{\frac{1 + \log|\overline{\mathcal{P}}|}{m^2 T}}$, we have $\mathbb{E}[R_T] \in \mathcal{O}\left(m\sqrt{T \log |\overline{\mathcal{P}}|}\right)$.*

# 6   Conclusion and Discussion

We designed revenue-optimal learning algorithms for pricing data. First, we leveraged properties like smoothness and diminishing returns to create novel discretization schemes for approximating any pricing curve. These schemes were then used in our learning algorithms to improve their statistical and computational properties. Our algorithms build on classical methods like UCB and FTPL but required significant adaptations to handle the vast space of pricing curves and the asymmetric feedback. An interesting future direction would be to relax the assumption that the seller knows the valuation curves $v_i$.

**Computational complexity.**   Our algorithm is designed to achieve polynomial computational complexity with respect to the number of data points when the number of types is fixed, making it suitable for practical data pricing scenarios where the type count is typically small or bounded. While the overall computational cost grows exponentially with the number of types due to the problem's strong NP-hardness (see [14]), this design choice ensures computational feasibility in settings with large datasets and a limited number of types.

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

# A   Omitted Details from Section 3

## A.1   Proof of Lemma 3.1

**Lemma 3.1.** *Assume there are $m$ types with non-decreasing value curves $\{v_i\}_{i\in[m]}$. For any non-decreasing price curve $p$, there exists an "$m$-step" price curve $\bar{p}$ that yields expected revenue at least that of $p$ with respect to any distribution over the $m$ types. Here, $m$-step refers to non-decreasing functions $f : [N] \to [0,1]$ where $f(n+1) - f(n) > 0$ in at most $m$ points (i.e., at most $m$ jumps).*

*Proof of Lemma 3.1.* Fix a price curve $p$. Let $n_{i,p}$ be the amount of data type $i$ purchase at price curve $p$, that is

$$n_{i,p} \triangleq \max \left\{ \underset{n \in [N]}{\mathrm{argmax}}(v_i(n) - p(n)) \right\}.$$

For $\{n_{i,p}\}_{i\in[m]}$, let $\pi : [m] \to [m]$ be a permutation such that $n_{\pi(1),p} \le n_{\pi(2),p} \le \cdots \le n_{\pi(m),p}$. Let $n_{(i)} \triangleq n_{\pi(i),p}$. Then, define a function $\bar{p} : [N] \to [0,1]$ as follows,

$$\bar{p}(n) \triangleq \begin{cases} p\left(n_{(1)}\right), & n \le n_{(1)}, \\ p\left(n_{(2)}\right), & n_{(1)} < n \le n_{(2)}, \\ \quad \vdots & \\ p\left(n_{(m-1)}\right), & n_{(m-2)} < n \le n_{(m-1)}, \\ p\left(n_{(m)}\right), & n_{(m-1)} < n \le N, \end{cases}$$

so that $\bar{p}$ has at most $m$ steps. Then, $\bar{p}$ has following properties,

$$\bar{p}(n) = p(n), \text{ when } n \in \left\{ n_{(1)}, n_{(2)}, \ldots, n_{(m)} \right\},$$
$$\bar{p}(n) \le p(n), \text{ when } n \in [N] \setminus \left\{ n_{(1)}, n_{(2)}, \ldots, n_{(m)} \right\}.$$

We next prove that for any $i \in [m]$, after changing the price function from $p$ to $\bar{p}$, the type $i$ buyer either purchases at $(n_{i,p}, p(n_{i,p}))$ or at $(N, p(n_{(m)}))$.

For any type $i$ and any amount of data $n \le n_{(m)}$, there exists $k$ such that $n_{(k-1)} < n \le n_{(k)}$ (let $n_{(0)} = 0$), we then have

$$v_i(n) - \bar{p}(n) \le v_i\left(n_{(k)}\right) - \bar{p}\left(n_{(k)}\right) \qquad \text{(as } v_i \text{ is non-decreasing and } \bar{p} \text{ is a step function.)}$$
$$= v_i\left(n_{(k)}\right) - p\left(n_{(k)}\right) \qquad\qquad \text{(as } \bar{p}\left(n_{(k)}\right) = p\left(n_{(k)}\right))$$

$$\leq v_i(n_{i,p}) - p(n_{i,p}) \qquad \qquad \text{(as } n_{i,p} \text{ maximizes the buyer's utility.)}$$
$$= v_i(n_{i,p}) - \bar{p}(n_{i,p}). \qquad \qquad \text{(as } \bar{p}(n_{i,p}) = p(n_{i,p}))$$

As shown in the above, type $i$ still prefers purchasing $n_{i,p}$ data over all $n \leq n_{(m)}$ under price $\bar{p}$.

For $n \in \{n_{(m)} + 1, \ldots, N\}$, by the monotonicity of value curves, we have

$$N = \max \left\{ \underset{n \in \{n_{(m)}+1,\ldots,N\}}{\arg\max} (v_i(n) - \bar{p}(n)) \right\}.$$

Therefore, for any $i \in [m]$, type $i$ either purchases at $(n_{i,p}, p(n_{i,p}))$, or purchases at $(N, \bar{p}(N)) = (N, p(n_{(m)}))$ under price $\bar{p}$. No matter in which case, type $i$ contributes no less revenue under $\bar{p}$ than $p$. It then follows that, for any type distribution $q$,

$$\text{rev}(\bar{p}) \geq \text{rev}(p).$$

$\square$

## A.2  Proof of Theorem 3.1

In this subsection, we prove Theorem 3.1 by decomposing it into three technical lemmas (Lemma A.1, A.2 and A.3). In Lemma A.1 and A.2, we prove the approximation guarantee of our discretization scheme and, in Lemma A.3 we provide an upper bound on the size of the discretization.

**Lemma A.1.** *For any type distribution, there exists a pricing function $\widetilde{p} : [N] \to [\epsilon, 1]$ such that*

$$\text{rev}(\widetilde{p}) \geq \text{OPT} - \epsilon.$$

*Proof of Lemma A.1.* Consider the optimal pricing function $p^\star : [N] \to [0, 1]$, i.e., $\text{OPT} = \text{rev}(p^\star)$. Consider price curve $\widetilde{p} : [N] \to [\epsilon, 1]$ where $\widetilde{p}(n) = \max(\epsilon, p^\star(n))$.

Let $J \triangleq \{n \in [N] : \widetilde{p}(n) = p^\star(n)\}$ be the set of data quantities whose price under $\widetilde{p}$ are the same as those under $p$. Any buyer type who would have purchased $n \in J$ amount of data under $p^\star$ will purchase the same amount of data under $\widetilde{p}$. On the other hand, for buyer types who would have purchased $n \notin J$ amount of data under $p^\star$, since $\widetilde{p}(n) = \epsilon > p^\star(n)$ for $n \notin J$, the expected revenue contribution from such buyers under $p^\star$ is at most $\epsilon$, hence no matter they purchase or not under $\widetilde{p}$, we have $\text{rev}(\widetilde{p}) \geq \text{OPT} - \epsilon$. $\square$

**Lemma A.2.** *For any $\widetilde{p} \in [\epsilon, 1]^N$ there exists $p' \in \overline{\mathcal{P}}$ such that $\text{rev}(p') \geq \text{rev}(\widetilde{p})/(1 + \epsilon)$, for any type distribution $q$.*

*Proof of Lemma A.2.* For $m$ buyer types, by Lemma 3.1, there exists a non-decreasing step function $\bar{p} \in [\epsilon, 1]^N$ with at most $m$ steps, whose expected revenue is at least $\text{rev}(\widetilde{p})$. Assume $\bar{p}$ has $k$ steps, $k \leq m$. To simplify the notation, for $1 \leq j \leq k$, let $\bar{p}_j$ denote the price $\bar{p}$ on $j$th step. That is,

$$\bar{p}(n) = \begin{cases} \bar{p}_1, & n \in (0, i_1] \cap \mathbb{Z}, \\ \bar{p}_2, & n \in (i_1, i_2] \cap \mathbb{Z}, \\ \quad \vdots \\ \bar{p}_k, & n \in (i_{k-1}, N] \cap \mathbb{Z}. \end{cases}$$

Where $i_1, \ldots, i_{k-1} \in [N]$ are discontinuities in $\bar{p}$.

Recall the definitions of $Z$ and $W$ as stated in Algorithm 1,

$$Z_i \triangleq \left\{ \epsilon(1+\epsilon)^i : \forall i \in \left\{0, 1, \ldots, \left\lceil \log_{1+\epsilon} \frac{1}{\epsilon} \right\rceil \right\} \right\}, \ Z = \bigcup_i Z_i.$$

$$W_i \triangleq \left\{ Z_{i-1} + Z_{i-1} \cdot \frac{\epsilon k}{m} : \forall k \in \{1, 2, \ldots, \lceil (2+\epsilon)m \rceil\} \right\}, \quad W \triangleq \bigcup_{i=1}^{\lceil \log_{1+\epsilon} \frac{1}{\epsilon} \rceil} W_i.$$

Let $i_k = N$ and for each $j \in [k]$, let $Z_{i_j}$ be the price obtained by rounding $\bar{p}_j$ down to the nearest value in $Z$. By constructions of $Z$ and $W$ above, $W_{i_j}$ is a partition of interval $(Z_{i_j-1}, Z_{i_j+1})$. Let $w_j$ be the price obtained by rounding $\bar{p}_j$ down to the nearest value in $W_{i_j}$. Set $d_j \triangleq \frac{\epsilon}{m} \cdot Z_{i_j-1}$ and consider $k$-step function $p$ defined by whose price at $j$th step (denoted $p_j$) is $w_j - (j-1)d_j \in W_{i_j}$, that is

$$
p(n) = \begin{cases}
p_1 = w_1, & \text{for } n \in (0, i_1] \cap \mathbb{Z}, \\
p_2 = w_2 - d_2, & \text{for } n \in (i_1, i_2] \cap \mathbb{Z}, \\
\quad \vdots \\
p_k = w_k - (k-1)d_k, & \text{for } n \in (i_{k-1}, N] \cap \mathbb{Z}.
\end{cases}
$$

By the tie-breaking rule and the monotonicity of valuation curves, buyers only purchase among $0, i_1, i_1, \ldots, i_k$ number of data under $p$ and $\bar{p}$.

*Subclaim.* Then, $p$ and $\bar{p}$ satisfies the following

$$
\text{rev}(p) \geq \text{rev}(\bar{p})/(1+\epsilon), \tag{8}
$$

with respect to any type distribution.

*Proof of the Subclaim.* We prove the above subclaim with two steps.

**Step 1**: No buyer who prefers to purchase $i_j$ data under $\bar{p}$ would prefer $i_{j'}$ data for some $j' < j$ under $p$ (i.e., one with a less price). This is because, when going from price $\bar{p}$ to $p$, the increase in the buyer's utility for $i_j$ data is $\bar{p}_j - p_j$, which is higher than the increase $\bar{p}_{j'} - p_{j'}$ for $i_{j'}$ data. Formally, this can be seen as follows: For any $j' < j$ we have,

$$
\bar{p}_j - p_j \geq w_j - p_j = (j-1)d_j,
$$

as $\bar{p}_j \geq w_j$ and $p_j = w_j - (j-1)d_j$. Moreover,

$$
\bar{p}_{j'} < w_{j'} + d_{j'} \implies \bar{p}_{j'} - p_{j'} < w_{j'} + d_{j'} - p_{j'} = j'd_{j'}. \tag{9}
$$

The inequality $\bar{p}_{j'} < w_{j'} + d_{j'}$ holds because $w_{j'}$ is the result of rounding down $\bar{p}_j$ to the nearest value in $W_{i_j}$.

By constructions of sets $Z$ and $W$, we have $d_j \geq d_{j'}$ which implies $(j-1)d_j \geq j'd_{j'}$. Then, by combining the above inequalities, we obtain

$$
\bar{p}_j - p_j \geq (j-1)d_j \geq j'd_{j'} \geq \bar{p}_{j'} - p_{j'}. \tag{10}
$$

Consider a buyer with value curve $v$ who prefers to purchase at $i_j$ under price $\bar{p}$, then it must be

$$
v(i_j) - \bar{p}_j > v(i_{j'}) - \bar{p}_{j'}. \tag{11}
$$

Then, by combining (10) and (11), we have

$$
v(i_j) - p_j > v(i_{j'}) - p_{j'},
$$

therefore the buyer would not purchase at $i_{j'} < i_j$ under $p$.

**Step 2**: Next, we claim that $p_j \geq \bar{p}_j/(1+\epsilon)$ for all step $j \in [k]$. Since $Z_{i_j}$ is obtained by rounding $\bar{p}_j$ down to the nearest value in $Z$, we have

$$
\bar{p}_j \geq Z_{i_j} = Z_{i_j-1} + \epsilon Z_{i_j-1} = Z_{i_j-1} + md_j. \tag{12}
$$

By (9) and the above, we have

$$
p_j \geq \bar{p}_j - jd_j \geq Z_{i_j-1} + (m-j)d_j \geq Z_{i_j-1},
$$

where the first inequality is by (9), the second is by (12), and the third is because $m \leq j$.

Then, it follows that

$$
\bar{p}_{j'} - p_j \leq j \cdot d_j = \epsilon \cdot \frac{j}{m} \cdot Z_{i_j-1} \leq \epsilon \cdot Z_{i_j-1} \leq \epsilon \cdot p_j \implies p_j \geq \bar{p}_j/(1+\epsilon).
$$

So far we have proved $p_j \geq \bar{p}_j/(1 + \epsilon)$ and no type wants to change their preference to a smaller amount of data under $p$. If one type purchase at $\bar{p}_i$ under $\bar{p}$ and $p_k$ under $p$ for $k \geq i$, then $p_k \geq p_i \geq \bar{p}_i/(1 + \epsilon)$. Therefore, we have

$$\text{rev}(p) \geq \text{rev}(\bar{p})/(1 + \epsilon) \geq \text{rev}(\tilde{p})/(1 + \epsilon).$$

Since the construction of price $p$ is not relevant to type distribution, the above holds for any type distribution $q$, which proves the subclaim. $\square$

Note that $p$ constructed in the above subclaim is not necessarily non-decreasing as a larger amount of data surfers more price deduction when going from $\bar{p}$ to $p$. In this case, we can directly construct a non-decreasing price curve $p' \in \overline{\mathcal{P}}$ from $p$ such that

$$\text{rev}(p') \geq \text{rev}(\bar{p})/(1 + \epsilon).$$

Let $S \triangleq \{i \in [k] : \exists j < i, \text{ s.t. } p_j > p_i\}$. If $S$ is empty, this implies that $p$ is non-decreasing, hence setting $p' = p$. If $S$ is not empty, we define $p'$ as follows: Let $p'$ be a $k$-step function with the same jump points $i_1, \ldots, i_k$ as $p$. Let $p'_i$ be the value of $p'$ on $i$th step. Then, for $i \notin S$, let $p'_i = p_i$; and for $i \in S$, let $p'_i = \max_{j \notin S, j < i} p_j$. By construction, $p'$ is non-decreasing. Moreover, $p' = p$ on set $S^c$ and $p' > p$ on set $S$.

Next, we claim that $\bar{p}_j - p'_j$ is non-decreasing for all $j \in [k]$. Both $(\bar{p}_j - p_j)_{j \in [k]}$ and $\bar{p}$ are non-decreasing with respect to $j$ by the previous results. Hence,

$$
\begin{aligned}
\bar{p}_j - p'_j &< \bar{p}_j - p'_j \leq \bar{p}_{j+1} - p_{j+1} = \bar{p}_{j+1} - p'_{j+1}, &&\text{if } j \in S, j+1 \notin S, \\
\bar{p}_j - p'_j &= \bar{p}_j - p'_j \leq \bar{p}_{j+1} - p_{j+1} = \bar{p}_{j+1} - p'_{j+1}, &&\text{if } j \notin S, j+1 \notin S, \\
\bar{p}_j - p'_j &= \bar{p}_j - p'_{j+1} \leq \bar{p}_{j+1} - p'_{j+1}, &&\text{if } j \notin S, j+1 \in S, &&(\text{as } p'_{j+1} = p'_j) \\
\bar{p}_j - p'_j &= \bar{p}_j - p'_{j+1} \leq \bar{p}_{j+1} - p'_{j+1}, &&\text{if } j \in S, j+1 \in S. &&(\text{as } p'_{j+1} = p'_j)
\end{aligned}
$$

Therefore, any type that prefers to purchase at $j$th step under $\bar{p}$ would not prefer purchasing at any step $j' < j$ under $p'$, and since $p'_j \geq p_j \geq \bar{p}_j/(1 + \epsilon)$, we have

$$\text{rev}(p') \geq \text{rev}(\bar{p})/(1 + \epsilon) \geq \text{rev}(\tilde{p})/(1 + \epsilon).$$

$\square$

**Lemma A.3.** *When $n > m$, $|\overline{\mathcal{P}}| \leq \left(\frac{eN}{m}\right)^m \left(e\lceil(2 + \epsilon)\rceil \lceil\log_{1+\epsilon} \frac{1}{\epsilon}\rceil\right)^m$.*

*Proof of Lemma A.3.* For any integer $i \leq m$, the number of non-decreasing $i$-step price function is $\binom{N-1}{i}\binom{|W|}{i}$, hence we have

$$
\begin{aligned}
|\overline{\mathcal{P}}| &= \sum_{i=1}^{m} \binom{N-1}{i}\binom{|W|}{i} \\
&\leq \left(\sum_{i=1}^{m}\binom{N-1}{i}\right)\left(\sum_{i=1}^{m}\binom{|W|}{i}\right) \\
&\leq \left(\sum_{i=0}^{m}\binom{N-1}{i}\right)\left(\sum_{i=0}^{m}\binom{|W|}{i}\right) \\
&\leq \left(\frac{e(N-1)}{m}\right)^m \left(\frac{e|W|}{m}\right)^m \\
&\leq \left(\frac{e(N-1)}{m}\right)^m \left(e\lceil(2 + \epsilon)\rceil \left\lceil\log_{1+\epsilon} \frac{1}{\epsilon}\right\rceil\right)^m
\end{aligned}
$$

In the last inequality, we use the fact that $|W| \leq \lceil(2 + \epsilon)m\rceil \lceil\log_{1+\epsilon} \frac{1}{\epsilon}\rceil$. $\square$

Finally, Theorem 3.1 follows directly from the above lemmas.

**Theorem 3.1.** *Consider the discretization $\overline{\mathcal{P}}$ as constructed in Algorithm 1. For any type distribution, there exists $p \in \overline{\mathcal{P}}$ such that $\mathrm{rev}(p) \geq \mathrm{OPT} - \mathcal{O}(\epsilon)$. Moreover, we have $|\overline{\mathcal{P}}| \leq \left(\frac{e(N-1)}{m}\right)^m \left(e \lceil (2+\epsilon) \rceil \lceil \log_{1+\epsilon} \frac{1}{\epsilon} \rceil \right)^m \in \widetilde{\mathcal{O}}\left(\left(\frac{N}{\epsilon}\right)^m\right).$*

*Proof of Theorem 3.1.* Combining Lemma A.1 and Lemma A.2 together, we conclude that there exists price curve $p' \in \overline{\mathcal{P}}$ such that

$$\mathrm{rev}(p') \geq \frac{\mathrm{rev}(\tilde{p})}{1+\epsilon} \geq \frac{\mathrm{OPT} - \epsilon}{1+\epsilon} \geq \mathrm{OPT} - \frac{2\epsilon}{1+\epsilon} = \mathrm{OPT} - \mathcal{O}(\epsilon).$$

The size of $\overline{\mathcal{P}}$ follows from Lemma A.3. □

## A.3 Price discretization scheme for smooth monotonic valuations

We study discretization schemes to approximate monotone valuations under the smoothness condition in Assumption 1. Our procedure is outlined in Algorithm 5. The discretization $W$ of the valuation space follows Algorithm 1. Additionally, we uniformly split the data space into multiples of $\lfloor \frac{\epsilon N}{mL} \rfloor$, denoting them as the set $N_{\mathbf{S}}$. We then set the discretization $\overline{\mathcal{P}}$ to be the class of all "$m$-step" price curves on the function space $N_{\mathbf{S}} \to W$. The following theorem, proven in Appendix A.4, outlines the main properties of this discretization scheme: the size of the discretization has no dependence on the number of data $N$.

---

**Algorithm 5** Price discretization scheme for smooth monotonic valuations

---

**Given:** Smoothness constant $L$, approximation parameter $\epsilon > 0$.
Let $W$ be discretization of the valuation space $[0,1]$ given in Algorithm 1.

Let $N_{\mathbf{S}}$ be the following discretization of the interval $[0, N]$,

$$\delta \triangleq \left\lfloor \frac{\epsilon N}{mL} \right\rfloor, \qquad N_{\mathbf{S}} \triangleq \left\{ \delta k : \ k \in \left\lceil \frac{N}{\delta} \right\rceil \right\}.$$

Set $\overline{\mathcal{P}}$ to be the class of all "$m$-step" functions mapping $N_{\mathbf{S}} \to W$.

---

## A.4 Proof of Theorem 3.2

**Theorem 3.2.** *Consider the discretization $\overline{\mathcal{P}}$ as constructed in Algorithm 5. Under Assumption 1, for any type distribution, there exists $p \in \overline{\mathcal{P}}$ such that $\mathrm{rev}(p) \geq \mathrm{OPT} - \mathcal{O}(\epsilon)$. Moreover, $|\overline{\mathcal{P}}| \in \mathcal{O}\left(\log_{1+\epsilon}^m (1/\epsilon) \cdot (L/\epsilon)^m\right) \in \widetilde{\mathcal{O}}\left(\left(\frac{L}{\epsilon^2}\right)^m\right).$*

*Proof of Theorem 3.2.* By Lemma 3.1, there is a revenue optimal price curve $p^\star : [N] \to [0,1]$ which is a $k$-step function, for some $k \in [m]$. Where $p^\star$ can be compactly represented as the following set of tuples:

$$\{(n_1^\star, p_1^\star), (n_2^\star, p_2^\star), \ldots, (n_k^\star, p_k^\star)\},$$

where $n_1^\star, \ldots, n_k^\star$ denote the locations of jumps and $p_i^\star$ denote the value of $p^\star$ on step $i \in [k]$ (i.e. $p^\star(n) = p_i^\star$ for $n \in (n_{i-1}^\star, n_i^\star]$).

Let $\bar{\epsilon} := \frac{\epsilon}{m}$. Next, we generate a price $p'$ using Algorithm 6, which ensures that the price curve $p$ generated in the following step (13) is non-decreasing. We demonstrate that in each round of Algorithm 6, we incur a revenue loss of at most $\bar{\epsilon}$. If $p_i' > p_{i-1}' + \bar{\epsilon}$, everything remains the same and thus does not affect the expected revenue. If not, we combine the price of step $i$ with step $i-1$, let $p_j' \triangleq p_j' - (p_i' - p_{i-1}')$ for $j = i, \ldots, k$. During this process, buyers either make purchases at the same step, or switch to purchase at a higher step. Note that $p_i' - p_{i-1}' < \bar{\epsilon}$, so the revenue loss of each type is at most $\bar{\epsilon}$. This implies that the revenue loss in each round is at most $\bar{\epsilon}$. As there are $k$ rounds, we lose expected revenue of at most $m\bar{\epsilon}$. We conclude that $\mathrm{rev}(p')$ is within a gap of $\epsilon$ from OPT, i.e., $\mathrm{rev}(p') \geq \mathrm{OPT} - \epsilon$.

---

**Algorithm 6** Auxiliary Price Adjustment

---

**Input:** Optimal price curve $p^\star$.
Let $p' = p^\star$.
**for** $i = 2, \ldots, k$ **do**
    **if** $p'_i < p'_{i-1} + \bar{\epsilon}$ **then**
        **for** $j = i, \ldots, k$ **do**
            $p'_j = p'_j - \left(p'_i - p'_{i-1}\right)$.
        **end for**
    **end if**
**end for**
**Output:** Price curve $p'$.

---

After combining some steps in Algorithm 6, Assume that $p'$ is a $\bar{k}$-step function ($\bar{k} \le k$) represented by

$$\left\{ (n'_1, p'_1), (n'_2, p'_2), \ldots, (n'_{\bar{k}}, p'_{\bar{k}}) \right\}.$$

Then, we define a new price curve $p \in \overline{\mathcal{P}}$ as follows: let $\delta := \left\lfloor \frac{\bar{\epsilon} N}{L} \right\rfloor$, then $p$ is a $\bar{k}$-step function represented by

$$\left\{ (n_1, p_1), (n_2, p_2), \ldots, (n_{\bar{k}}, p_{\bar{k}}) \right\},$$

where

$$n_i \triangleq \left\lfloor \frac{n'_i}{\delta} \right\rfloor \delta, \quad p_i \triangleq p'_i - i\bar{\epsilon}. \tag{13}$$

First, we show that no buyer who purchases at step $i$ under $p'$ would purchase at step $j < i$ under $p$. Let the buyer's valuation be $v$. First, we prove that the buyer's utility is non-negative at $n_i$:

$$
\begin{aligned}
v(n_i) - p_i &\ge v(n'_i) - \delta \cdot \frac{L}{N} - p_i && \text{(by } L/N\text{-Smoothness of } v.) \\
&= v(n'_i) - \delta \cdot \frac{L}{N} - p'_i + i\bar{\epsilon} \\
&\ge v(n'_i) - \bar{\epsilon} - p'_i + i\bar{\epsilon} && \text{(as } \delta \cdot \tfrac{L}{N} \le \tfrac{L}{N} \cdot \tfrac{\bar{\epsilon} N}{L} = \bar{\epsilon}.) \\
&= v(n'_i) - p'_i + (i-1)\bar{\epsilon} \\
&\ge v(n'_i) - p'_i \\
&\ge 0.
\end{aligned}
$$

Then, we prove that the buyer's utility at $n_i$ is larger than that of $n_j$ for $j < i$, therefore, the buyer would not prefer buying at step $j < i$ under price $p$.

$$
\begin{aligned}
v(n_i) - p_i - (v(n_j) - p_j) &\ge v(n'_i) - \delta \cdot \frac{L}{N} - v(n'_j) - (p_i - p_j) && \text{(by } L/N\text{-Smoothness of } v.) \\
&= v(n'_i) - \delta \cdot \frac{L}{N} - v(n'_j) - (p'_i - p'_j - (i-j)\bar{\epsilon}) \\
&\ge v(n'_i) - \bar{\epsilon} - v(n'_j) - (p'_i - p'_j - (i-j)\bar{\epsilon}) \ \text{(as } \delta \cdot \tfrac{L}{N} \le \tfrac{L}{N} \cdot \tfrac{\bar{\epsilon} N}{L} = \bar{\epsilon}) \\
&= (v(n'_i) - p'_i) - (v(n'_j) - p'_j) + (i-j-1)\bar{\epsilon} \\
&\ge (v(n'_i) - p'_i) - (v(n'_j) - p'_j) && \text{(as } i > j) \\
&\ge 0. && \text{(as the buyer prefers } n_i \text{ than } n_k \text{ under } p'.)
\end{aligned}
$$

Finally, fix the type distribution $(q_1, \ldots, q_m)$, then we have

$$\mathrm{rev}(p') - \mathrm{rev}(p) \le \sum_{h=1}^{m} q_h \left( \sum_{i=1}^{k} (p'_i - p_i) \cdot \mathbb{I}(\text{Type } j \text{ purchase at } p'_i \text{ under price } p') \right)$$

$$\leq m\bar{\epsilon}$$
$$= \epsilon. \qquad\qquad\qquad \text{(as } \epsilon = m\bar{\epsilon}.)$$

Hence, $\mathrm{rev}(p)$ is within a gap of $2\epsilon$ from OPT.

We then apply Theorem 3.1 to price $p$. Therefore, it is enough to consider price functions from the set $N_{\mathbf{S}} \triangleq \left\{ k\delta : k = 1, \ldots, \left\lceil \frac{N}{\delta} \right\rceil \right\} \subseteq [N]$ to $W$ to approximate the revenue within $\mathcal{O}(\epsilon)$ gap. Moreover, this discretization is of the size $\left\lceil \frac{N}{\delta} \right\rceil^{|W|} \in \mathcal{O}\left( \left( \log_{1+\epsilon}\left(\frac{1}{\epsilon}\right) \right)^m \left(\frac{L}{\epsilon}\right)^m \right)$ as $\left\lceil \frac{N}{\delta} \right\rceil \in \mathcal{O}\left(\frac{Lm}{\epsilon}\right)$. $\qquad\square$

## A.5 Proof of Theorem 3.3

**Theorem 3.3.** *Consider the discretization $\overline{\mathcal{P}}$ as constructed in Algorithm 2. Under Assumption 2, for any type distribution, there exists $p \in \overline{\mathcal{P}}$ such that $\mathrm{rev}(p) \geq \mathrm{OPT} - \mathcal{O}(\epsilon)$. Moreover,*

$$|\overline{\mathcal{P}}| \in \mathcal{O}\left( \left(\frac{J}{\epsilon^2}\right)^m \log^m\left(\frac{N\epsilon^2}{Jm}\right) \cdot \left(\log_{1+\epsilon}^m 1/\epsilon\right) \right) \in \widetilde{\mathcal{O}}\left( \left(\frac{J}{\epsilon^3}\right)^m \right).$$

*Proof of Theorem 3.3.* For each $i = 0, 1, \ldots, \left\lceil \log_{1+\epsilon^2}\left(\frac{N\epsilon^2}{2Jm}\right) \right\rceil$, let $Y_i \triangleq \left\lfloor \frac{2Jm}{\epsilon^2}(1+\epsilon^2)^i \right\rfloor$, and $Q_i$ be the set $\left\{ \left\lfloor Y_i + \frac{Y_i\epsilon^2}{2Jm}k \right\rfloor : k = 1, \ldots, \lfloor 2Jm \rfloor \right\}$, i.e., $Q_i$ splits the interval $[Y_i, Y_{i+1}]$ equally into $2mJ$ parts.

The union of $Q_i$s and the set $\left\{ 1, 2, \ldots, \left\lfloor \frac{2Jm}{\epsilon^2} \right\rfloor \right\}$ form a set of grids on $[0, N]$, denoted by $N_{\mathbf{D}}$. There are at most $\frac{2Jm}{\epsilon^2} + 2Jm \log_{1+\epsilon^2}\left(\frac{N\epsilon^2}{2Jm}\right)$ grids in total.

By Lemma 3.1, there is a revenue optimal price curve $p^\star : [N] \to [0, 1]$ which is a $k$-step function, for some $k \in [m]$. Where $p^\star$ can be compactly represented as the following set of tuples:

$$\left\{ (n_1^\star, p_1^\star), (n_2^\star, p_2^\star), \ldots, (n_k^\star, p_k^\star) \right\},$$

where $n_1^\star, \ldots, n_k^\star$ denote the locations of jumps and $p_i^\star$ denote the value of $p^\star$ on step $i \in [k]$ (i.e. $p^\star(n) = p_i^\star$ for $n \in (n_{i-1}^\star, n_i^\star]$).

Then, define a new $k$-step price curve $p$ via

$$\left\{ (n_1, p_1), (n_2, p_2), \ldots, (n_k, p_k) \right\},$$

where $n_i$ is given by

$$n_i \leftarrow \text{round down } n_i^\star \text{ to the closest grid in } N_{\mathbf{D}}.$$

Then we define $p_i$ below. If $p_i^\star < \epsilon(1 + \epsilon)$, let $p_i = \epsilon(1 + \epsilon)$; otherwise, let $Z_{n_i^\star}$ be the price obtained by rounding $p_i^\star$ down to the nearest value in $Z$. By constructions of $Z$ and $W$ above, $W_{n_i^\star}$ is a partition of interval $(Z_{n_i^\star-1}, Z_{n_i^\star+1})$. Let $w_i$ be the price obtained by rounding $p_i^\star$ down to the nearest value in $W_{n_i^\star}$. Set $d_i \triangleq \frac{\epsilon}{m} \cdot Z_{n_i^\star-1}$. Then define $p_i \triangleq w_i - i \cdot d_i \in W_{n_i^\star}$.

First, we prove for $i$ satisfying $p_i^\star > \epsilon(1 + \epsilon)$, if a buyer purchases at $n_i$ under price $p^\star$, she will not purchase at $n_j$, $j < i$ under new price $p$. We prove this property separately when $n_i \leq \frac{2Jm}{\epsilon^2}$ and $n_i > \frac{2Jm}{\epsilon^2}$.

(i) When $n_i > \frac{2Jm}{\epsilon^2}$.

The buyer's utility at $n_i$ under price $p$ is,

$$v(n_i) - p_i = v(n_j^\star) - p_i^\star + \left( p_i^\star - p_i - (v(n_i^\star) - v(n_i)) \right). \qquad (14)$$

Let $\delta_i \triangleq v(n_i^\star) - v(n_i)$. Then $\delta_i$ is upper bounded by,

$$\delta_i = \sum_{h=n_i}^{n_i^\star-1} v(h+1) - v(h) \leq \sum_{h=n_i}^{n_i^\star-1} \frac{J}{h} \leq \frac{J}{n_i}(n_i^\star - n_i)$$

$$\leq \frac{J}{n_i} \cdot \left( n_i \cdot \frac{\epsilon^2}{2mJ} + 1 \right) = \frac{\epsilon^2}{2m} + \frac{J}{n_i} \leq \frac{\epsilon^2}{2m} + \frac{\epsilon^2}{2m} = \frac{\epsilon^2}{m}, \tag{15}$$

where the third inequality is due to Lemma A.4.

By the construction of $p$, we have

$$p_i^\star - p_i = Z_{n_i-1} \cdot \frac{\epsilon i}{m} \geq \frac{\epsilon^2 i}{m} \geq \frac{\epsilon^2}{m} \geq \delta_i. \tag{16}$$

Therefore, by (14), $v(n_i) - p_i \geq v(n_i^\star) - p_i^\star \geq 0$, buyer's utility at $n_i$ under price $p$ is non-negative.

Next, we claim that $v(n_i) - p_i - (v(n_j) - p_j) \geq 0$. To prove this, for any $j < i$, let $\delta_j \overset{\triangle}{=} v(n_j^\star) - v(n_j)$, then we have

$$v(n_i) - p_i - (v(n_j) - p_j)$$
$$= v(n_i^\star) - p_i^\star - (v(n_j^\star) - p_j^\star) + (p_i^\star - p_i - \delta_i) - (p_j^\star - p_j - \delta_j)$$

Where $v(n_i^\star) - p_i^\star - (v(n_j^\star) - p_j^\star) \geq 0$ because the buyer prefers $n_i^\star$ over $n_j^\star$ under price $p^\star$. Recall that we have $\delta_j \geq 0$, then we bound $\delta_i - \delta_j$ as follows,

$$\delta_i - \delta_j \leq \delta_i \leq \frac{\epsilon^2}{m}. \tag{17}$$

By the construction of $p_i$, we have,

$$p_i^\star - p_i - (p_j^\star - p_j) = Z_{n_i-1} \cdot \frac{\epsilon i}{m} - Z_{n_j-1} \cdot \frac{\epsilon j}{m}$$
$$\geq Z_{n_j-1} \cdot \left( \frac{\epsilon i}{m} - \frac{\epsilon j}{m} \right) \qquad \text{(as } Z_{n_i-1} \geq Z_{n_j-1}\text{)}$$
$$\geq Z_{n_j-1} \cdot \left( \frac{\epsilon}{m} \right) \qquad \text{(as } i > j\text{)}$$
$$\geq \frac{\epsilon^2}{m}. \tag{18}$$

Therefore, combining (17) and (18) together, we have

$$v(n_i) - p_i - (v(n_j) - p_j) \geq v(n_i^\star) - p_i^\star - (v(n_j^\star) - p_j^\star) \geq 0.$$

We conclude that under price $p$, the buyer prefers $n_i$ over $n_j$, for any $j < i$.

(ii) When $n_i \leq \frac{2Jm}{\epsilon^2}$.

In this case, $n_i = n_i^\star$, and for any $j < i$, we still have $n_j = n_j^\star$. First, we prove the buyer's utility at $n_i'$ under $p$ is non-negative:

$$v(n_i) - p_i = v(n_i^\star) - p_i$$
$$= v(n_i^\star) - p_i^\star + (p_i^\star - p_i)$$
$$\geq v(n_i^\star) - p_i^\star$$
$$\geq 0.$$

Then, we show that the buyer prefers $n_i$ over $n_j$ under $p$:

$$v(n_i) - p_i - (v(n_j) - p_j) = v(n_i^\star) - p_i^\star - (v(n_j^\star) - p_j^\star) + (p_i^\star - p_i - \delta_i) - (p_j^\star - p_j - \delta_j)$$
$$= v(n_i^\star) - p_i^\star - (v(n_j^\star) - p_j^\star) + (p_i^\star - p_i) - (p_j^\star - p_j)$$
$$\geq v(n_i^\star) - p_i^\star - (v(n_j^\star) - p_j^\star)$$
$$\geq 0,$$

where the first inequality is due to (18), and the second is because the buyer prefers $n_i^\star$ over $n_j^\star$ under $p^\star$.

So far we have completed the proof that for $i$ satisfying $p_i^\star > \epsilon(1 + \epsilon)$, if a buyer purchases at $n_i$ under price $p^\star$, she will not purchase at $n_j$, $j < i$ under new price $p$.

Then, similar to Step 2 in the proof of Lemma A.2, we have $p \geq \frac{p^\star}{1+\epsilon}$ pointwise. We then conclude the proof by observing

$$\text{rev}(p) \geq \frac{\text{rev}(p^\star) - \mathcal{O}(\epsilon)}{1 + \epsilon} = \text{OPT} - \mathcal{O}(\epsilon).$$

□

**Lemma A.4.** When $n_i > \frac{2Jm}{\epsilon^2}$, we have $n_j^\star - n_i \leq n_i \cdot \frac{\epsilon^2}{2Jm} + 1$.

*Proof of Lemma A.4.* By the construction of discretization set, $n_i$ must have the following form,

$$\left\lfloor Y_{i'} + Y_{i'} \cdot \frac{\epsilon^2 k'}{2Jm} \right\rfloor, \text{ where } Y_{i'} = \left\lfloor \frac{2Jm}{\epsilon^2} (1 + \epsilon^2)^{i'} \right\rfloor \text{ for some } i', k' \in \mathbb{Z}.$$

Since $n'_j$ is obtained by rounding down $n_j$ to the nearest grid in $N_{\mathbf{D}}$, $n_j$ satisfies the following inequality,

$$n_j \leq n_j^\star \leq Y_{i'} + Y_{i'} \cdot \frac{\epsilon^2(k'+1)}{2Jm}.$$

Therefore, we have

$$\begin{aligned}
n_i^\star - n_i &\leq Y_{i'} + Y_{i'} \cdot \frac{\epsilon^2(k'+1)}{2Jm} - n_i \\
&= Y_{i'} + Y_{i'} \cdot \frac{\epsilon^2(k'+1)}{2Jm} - \left\lfloor Y_{i'} + Y_{i'} \cdot \frac{\epsilon^2 k'}{2Jm} \right\rfloor \\
&\leq Y_{i'} + Y_{i'} \cdot \frac{\epsilon^2(k'+1)}{2Jm} - \left( Y_{i'} + Y_{i'} \cdot \frac{\epsilon^2 k'}{2Jm} \right) + 1 \\
&= Y_{i'} \cdot \frac{\epsilon^2}{2Jm} + 1 \\
&\leq n_i \cdot \frac{\epsilon^2}{2Jm} + 1.
\end{aligned}$$

Where in the last inequality, since $Y_{i'}$ is an integer, and we have

$$n'_i = \left\lfloor Y_{i'} + Y_{i'} \cdot \frac{\epsilon^2 k'}{2Jm} \right\rfloor \geq Y_{i'}, \text{ for } k' \geq 0.$$

□

# B  Proof of Theorem 5.1

**Theorem 5.1.** *Suppose in Algorithm 4 we use a discretization $\overline{\mathcal{P}}$ which is a $\mathcal{O}(1/\sqrt{T})$ additive approximation to any price curve. Let $R_T$ be as defined in (4). Then, for Algorithm 4, we have $\mathbb{E}[R_T] \in \mathcal{O}\left(m^2\theta T + \theta^{-1}\left(1 + \log|\overline{\mathcal{P}}|\right)\right)$. Setting $\theta = \sqrt{\frac{1+\log|\overline{\mathcal{P}}|}{m^2 T}}$, we have $\mathbb{E}[R_T] \in \mathcal{O}\left(m\sqrt{T\log|\overline{\mathcal{P}}|}\right)$.*

*Proof of Theorem 5.1.* Recall that the regret $R_T$ for the adversarial setting is

$$\begin{aligned}
R_T &\stackrel{\Delta}{=} \max_{p \in \mathcal{P}} \sum_{t=1}^{T} r(i_t, p) - \sum_{t=1}^{T} r(i_t, p_t) \\
&= \underbrace{\max_{p \in \mathcal{P}} \sum_{t=1}^{T} r(i_t, p) - \max_{p \in \overline{\mathcal{P}}} \sum_{t=1}^{T} r(i_t, p)}_{\text{Loss of revenue due to discretization}} + \underbrace{\max_{p \in \overline{\mathcal{P}}} \sum_{t=1}^{T} r(i_t, p) - \sum_{t=1}^{T} r(i_t, p_t)}_{\stackrel{\Delta}{=} \overline{R}_T \text{ (discretization regret)}}. \quad (19)
\end{aligned}$$

We decompose $R_T$ into two regrets. The first term is the sacrifice of revenue on discretization. The second term is the algorithm regret when competing against the optimal price within the discretization set $\overline{\mathcal{P}}$.

According to Theorem 3.1, our discretization scheme approaches optimal revenue within a gap of $\frac{2\epsilon}{1+\epsilon}$:

$$\max_{p \in \mathcal{P}} \sum_{t=1}^{T} r(i_t, p) - \max_{p \in \overline{\mathcal{P}}} \sum_{t=1}^{T} r(i_t, p) \leq \frac{2\epsilon T}{1 + \epsilon} < 2\epsilon T. \tag{20}$$

Therefore, the first term can be bounded by $2\epsilon T$.

According to Theorem B.1, the second term discretization regret is upper bounded by

$$\mathbb{E}[\overline{R}_T] \leq 3m \sqrt{T \log |\overline{\mathcal{P}}|}. \tag{21}$$

Combining (20) and (21) together, we have,

$$\mathbb{E}[R_T] \leq 2\epsilon T + 3m \sqrt{T \log |\overline{\mathcal{P}}|} = \mathcal{O}\left( m \sqrt{T \log |\overline{\mathcal{P}}|} \right). \qquad \text{(as } \epsilon = \tfrac{1}{\sqrt{T}})$$

Plug in the size of discretization set in Section 3, we have,

$$\mathbb{E}[R_T] = \widetilde{\mathcal{O}}\left( m^{3/2} \sqrt{T} \right).$$

$\square$

**Theorem B.1.** *The discretization regret $\overline{R}_T$ defined in* (19) *has upper bound* $\mathcal{O}\left( m \sqrt{T \log |\overline{\mathcal{P}}|} \right)$.

*Proof of Theorem B.1.* We first claim that $r_t(p_t) = r(i_t, p_t)$ all $t$. If the buyer make a purchase at round $t$, $r_t(p_t) = r(i_t, p_t)$ holds by definition. But if the buyer does not purchase at a price $p_t$ on round $t$, $r(i_t, p_t) = 0$. Since $S_t^c$ contains all the types that would not make a purchase at $p_t$, we have $r(i, p_t) = 0, \forall i \in S_t^c$, and

$$r(i_t, p_t) = \sum_{i \in S_t^c} r(i, p_t) = r_t(p_t) = 0.$$

Therefore, $r_t(p_t) = r(i_t, p_t)$ holds for every round $t \in [T]$. Denote $p^\star$ as,

$$p^\star = \operatorname*{argmax}_{p \in \overline{\mathcal{P}}} \sum_{t=1}^{T} r(i_t, p).$$

Then, we decompose the regret as follows,

$$\begin{aligned}
\mathbb{E}[R_T] &= \sum_{t=1}^{T} r(i_t, p^\star) - \mathbb{E}\left[ \sum_{t=1}^{T} r(i_t, p_t) \right] \\
&= \sum_{t=1}^{T} r(i_t, p^\star) - \mathbb{E}\left[ \sum_{t=1}^{T} r_t(p_t) \right] \\
&= \mathbb{E}\left[ \sum_{t=1}^{T} (r(i_t, p^\star) - r_t(p^\star)) \right] + \mathbb{E}\left[ \sum_{t=1}^{T} r_t(p^\star) - \sum_{t=1}^{T} r_t(p_{t+1}) \right] + \mathbb{E}\left[ \sum_{t=1}^{T} r_t(p_{t+1}) - r_t(p_t) \right].
\end{aligned} \tag{22}$$

We bound three terms in (22) separately.

**The first term.** For any price $p$ and any round $t$, we have $r_t(p) \geq r(i_t, p)$ by definition. Hence,

$$\sum_{t=1}^{T} (r(i_t, p^\star) - r_t(p^\star)) \leq 0. \tag{23}$$

**The second term.** Since $p^\star = \operatorname*{argmax}_{p \in \overline{\mathcal{P}}} \sum_{t=1}^{T} r(i_t, p)$. We apply Lemma B.1 to $p^\star$,

$$\sum_{t=1}^{T} r_t(p^\star) - \sum_{t=1}^{T} r_t(p_{t+1}) \leq \theta_{p_1} - \theta_{p^\star}.$$

Note that both $\theta_{p_1}$ and $\theta_{p^*}$ are drawn i.i.d. from exponential distribution,

$$\mathbb{E}[\theta_{p_1}] \leq \mathbb{E}\left[\max_{p \in \overline{\mathcal{P}}} \theta_p\right] \leq \frac{1 + \log|\overline{\mathcal{P}}|}{\theta},$$

$$\mathbb{E}[\theta_{p^*}] \leq \mathbb{E}\left[\max_{p \in \overline{\mathcal{P}}} \theta_p\right] \leq \frac{1 + \log|\overline{\mathcal{P}}|}{\theta}.$$

We have

$$\mathbb{E}\left[\sum_{t=1}^{T} r_t(p^*) - \sum_{t=1}^{T} r_t(p_{t+1})\right] \leq \mathbb{E}\left[\theta_{p_1} - \theta_{p^*}\right] \leq \frac{1 + \log|\overline{\mathcal{P}}|}{\theta}. \tag{24}$$

**The third term.** Note that for any price $p \in \overline{\mathcal{P}}$ and any round $t$, $r_t(p) \leq m$. Therefore we have,

$$\mathbb{E}\left[r_t(p_{t+1}) - r_t(p_t)\right] = \mathbb{P}\left(p_{t+1} \neq p_t\right) \mathbb{E}\left[r_t(p_{t+1}) - r_t(p_t) \mid p_{t+1} \neq p_t\right] \leq m \cdot \mathbb{P}\left(p_{t+1} \neq p_t\right).$$

The price curve on round $t$ is $p_t$, then by the price updation rule,

$$p_t = \operatorname*{argmax}_{p \in \overline{\mathcal{P}}} \sum_{\tau=1}^{t-1} r_\tau(p) + \theta_p,$$

which is equivalent to,

$$\theta_{p_t} \geq \theta_p + \sum_{\tau=1}^{t-1} r_\tau(p) - \sum_{\tau=1}^{t-1} r_\tau(p_t), \ \forall p \in \overline{\mathcal{P}}.$$

For all $p' \in \overline{\mathcal{P}}$, let $c_{t-1,p'}$ denote

$$\max_{p \in \overline{\mathcal{P}}} \left(\theta_p + \sum_{\tau=1}^{t-1} r_\tau(p) - \sum_{\tau=1}^{t-1} r_\tau(p')\right) \triangleq c_{t-1,p'}, \tag{25}$$

then $p_t = p'$ is equivalent to

$$\theta_{p'} \geq c_{t-1,p'}. \tag{26}$$

**Subclaim.** If $\theta_{p_t}$ also satisfies the following condition (27),

$$\theta_{p_t} \geq \theta_p + \sum_{\tau=1}^{t-1} r_\tau(p) - \sum_{\tau=1}^{t-1} r_\tau(p_t) + m, \ \forall p \in \overline{\mathcal{P}}, \tag{27}$$

then $p_{t+1} = p_t$.

*Proof of the Subclaim.* If (27) holds for all $p \in \overline{\mathcal{P}}$,

$$\theta_{p_t} \geq \theta_p + \sum_{\tau=1}^{t-1} r_\tau(p) - \sum_{\tau=1}^{t-1} r_\tau(p_t) + m$$

$$\geq \theta_p + \sum_{\tau=1}^{t} r_\tau(p) - \sum_{\tau=1}^{t} r_\tau(p_t). \qquad \text{(because } \forall p \in \overline{\mathcal{P}}, r_t(p) \in [0, m])$$

Hence,

$$p_t = \operatorname*{argmax}_{p \in \overline{\mathcal{P}}} \sum_{\tau=1}^{t} r_\tau(p) + \theta_p = p_{t+1}.$$

$\square$

Therefore, (27) is a sufficient condition for $p_{t+1} = p_t$. We then bound the probability of $p_{t+1} = p_t$ by computing the probability of (27) happening.

$$\mathbb{P}\left(p_t = p_{t+1}\right) = \sum_{p \in \overline{\mathcal{P}}} \mathbb{P}\left(p_t = p\right) \mathbb{P}(p_{t+1} = p \mid p_t = p)$$

$$= \sum_{p \in \overline{\mathcal{P}}} \mathbb{P}\left(p_t = p\right) \mathbb{P}\left(p_{t+1} = p \mid \theta_p \geq c_{t-1,p}\right) \qquad (\text{ by } (26))$$

$$\geq \sum_{p \in \overline{\mathcal{P}}} \mathbb{P}\left(p_t = p\right) \mathbb{P}\left(\theta_p \geq c_{t-1,p} + m \mid \theta_p \geq c_{t-1,p}\right)$$

$$\geq \sum_{p \in \overline{\mathcal{P}}} \mathbb{P}\left(p_t = p\right) e^{-m\theta}$$

$$= e^{-m\theta}$$

$$\geq 1 - m\theta$$

Therefore, $\mathbb{P}\left(p_t \neq p_{t+1}\right) \leq m\theta$. Hence, the third term can be bounded as

$$\mathbb{E}\left[r_t(p_{t+1}) - r_t(p_t)\right] \leq m^2\theta \implies \sum_{t=1}^{T} \mathbb{E}\left[r_t(p_{t+1}) - r_t(p_t)\right] \leq m^2\theta T. \qquad (28)$$

Set $\theta = \sqrt{\frac{\log |\overline{\mathcal{P}}|}{m^2 T}}$. Combining the upper bounds for three terms (23), (24) and (28) together, we have

$$\mathbb{E}[R_T] \leq \frac{1 + \log |\overline{\mathcal{P}}|}{\theta} + m^2\theta T \in \mathcal{O}\left(m\sqrt{T \log |\overline{\mathcal{P}}|}\right).$$

Plugging in the size of the discretization set (Theorem 3.1), we have,

$$\mathbb{E}[R_T] \in \widetilde{\mathcal{O}}\left(m^{3/2}\sqrt{T}\right).$$

$\square$

**Lemma B.1.** *For any $p \in \overline{\mathcal{P}}$,*

$$\sum_{t=1}^{T} r_t(p_{t+1}) + \theta_{p_1} \geq \sum_{t=1}^{T} r_t(p) + \theta_p. \qquad (29)$$

*Proof of Lemma B.1.* We prove this by induction. For $T = 0$, the inequality $\theta_{p_1} \geq \theta_p$ holds by definition $p_1 = \underset{p \in \overline{\mathcal{P}}}{\arg\max}\, \theta_p$. Assume that the inequality holds for some $T$. Then for any $p \in \overline{\mathcal{P}}$,

$$\sum_{t=1}^{T+1} r_t(p_{t+1}) + \theta_{p_1} = \sum_{t=1}^{T} r_t(p_{t+1}) + \theta_{p_1} + r_{T+1}(p_{T+2})$$

$$\geq \sum_{t=1}^{T} r_t(p_{T+2}) + \theta_{p_{T+2}} + r_{T+1}(p_{T+2})$$

$$= \sum_{t=1}^{T+1} r_t(p_{T+2}) + \theta_{p_{T+2}}$$

$$\geq \sum_{t=1}^{T+1} r_t(p) + \theta_p.$$

Where the first inequality is by the induction hypothesis, and the second inequality is by

$$p_{T+2} = \underset{p \in \overline{\mathcal{P}}}{\arg\max} \sum_{t=1}^{T+1} r_t(p) + \theta_p.$$

By the induction, the inequality (29) holds for any $T \geq 0$. $\square$

# C Proof of Theorem 4.1

In this section, we prove, Theorem 4.1, our regret upper bound of Algorithm 3. We prove the theorem by first decomposing the regret into two parts: Regret with respect to the best price in a discretized set (called "discretization regret") and the residual error due to discretization. The residual error is controlled by the approximation guarantees developed in Section 3. Then, the key lemma in this appendix is Lemma C.1 which controls the discretization. We prove Lemma C.1 using a technique adapted from Chen et al. [15].

**Theorem 4.1.** *Suppose in Algorithm 3 we use a discretization $\overline{\mathcal{P}}$ which is a $\mathcal{O}(1/\sqrt{T})$ additive approximation to any price curve. Then, the regret of Algorithm 3 satisfies $\mathbb{E}[R_T] \in \widetilde{\mathcal{O}}(m\sqrt{T})$.*

*Proof of Theorem 4.1.* For the sake of simplicity, we define $r(i, p)$ as the revenue under type $i$ and price $p$, i.e, $r(i, p) \triangleq p(n_{i,p})$. Therefore, on every round, we have $r(i_t, p_t) = p_t(n_{i_t, p_t})$.

Recall that the regret $R_T$ is

$$
\begin{aligned}
R_T &\triangleq T \cdot \text{OPT} - \sum_{t=1}^{T} p_t(n_{i_t, p_t}) \\
&= T \cdot \text{OPT} - \sum_{t=1}^{T} r(i_t, p_t) \\
&= \underbrace{T \cdot \text{OPT} - T \cdot \max_{p \in \overline{\mathcal{P}}} \text{rev}(p)}_{\text{Loss of revenue due to discretization}} + \underbrace{T \cdot \max_{p \in \overline{\mathcal{P}}} \text{rev}(p) - \sum_{t=1}^{T} r(i_t, p_t)}_{\triangleq \ \overline{R}_T \ \text{(discretization regret)}}.
\end{aligned}
\tag{30}
$$

We decompose $R_T$ into two parts. The first term is the sacrifice of revenue on discretization. The second term is the algorithm regret when competing against the optimal price within the discretization set $\overline{\mathcal{P}}$.

According to Theorem 3.1, our discretization scheme approaches OPT within a gap of $\frac{2\epsilon}{1+\epsilon}$,

$$
\text{OPT} - \max_{p \in \overline{\mathcal{P}}} \text{rev}(p) \leq \frac{2\epsilon}{1 + \epsilon} \leq 2\epsilon.
$$

Therefore, the first term can be bounded as,

$$
T \cdot \text{OPT} - T \cdot \max_{p \in \overline{\mathcal{P}}} \text{rev}(p) \leq 2\epsilon T.
\tag{31}
$$

By Lemma C.1, the second term, discretization regret, is upper bounded by

$$
\mathbb{E}[\overline{R}_T] \leq 93m\sqrt{T \log T}
\tag{32}
$$

Combining (31) and (32) together, we have,

$$
\mathbb{E}[R_T] \leq 2\epsilon T + 93m\sqrt{T \log T} = \widetilde{\mathcal{O}}(m\sqrt{T}) \qquad (\text{ as } \epsilon = \tfrac{1}{\sqrt{T}})
$$

$\square$

**Lemma C.1.** *The discretization regret $\overline{R}_T$ defined in (30) is at most $\widetilde{\mathcal{O}}(m\sqrt{T})$.*

*Proof of Lemma C.1.* The discretization regret $\overline{R}_T$

$$
\begin{aligned}
\mathbb{E}[\overline{R}_T] &= \mathbb{E}\left[ T \cdot \max_{p \in \overline{\mathcal{P}}} \text{rev}(p) - \sum_{t=1}^{T} r(i_t, p_t) \right] \\
&= \mathbb{E}\left[ \sum_{t=1}^{T} (r(p^\star, i_t) - r(p_t, i_t)) \right]
\end{aligned}
$$

$$= \sum_{t=1}^{T} \mathbb{E}\left[r(p^\star, i_t) - r(p_t, i_t)\right]$$

$$= \sum_{t=1}^{T} \mathbb{E}\left[\text{rev}(p^\star) - \text{rev}(p_t)\right]$$

$$= \sum_{t=1}^{T} \mathbb{E}\left[(\text{rev}(p^\star) - \text{rev}(p_t)) \cdot \mathbb{I}(A_t)\right] + \sum_{t=1}^{T} \mathbb{E}\left[(\text{rev}(p^\star) - \text{rev}(p_t)) \cdot \mathbb{I}(A_t^c)\right]$$

$$\triangleq \sum_{t=1}^{T} \mathbb{E}\left[\delta_{p_t} \cdot \mathbb{I}(A_t)\right] + \sum_{t=1}^{T} \mathbb{E}\left[\delta_{p_t} \cdot \mathbb{I}(A_t^c)\right]. \tag{33}$$

We can further decompose $\mathbb{E}[\overline{R}_T]$ into $\sum_{t=1}^{T} \mathbb{E}\left[\delta_{p_t} \cdot \mathbb{I}(A_t)\right]$ and $\sum_{t=1}^{T} \mathbb{E}\left[\delta_{p_t} \cdot \mathbb{I}(A_t^c)\right]$. Where for any round $t$, we define the good event $A_t$ as follows,

$$\forall i \in [m], \quad q_i \le \widehat{q}_{i,t} \le q_i + 2\sqrt{\frac{\log T}{T_{i,t}}}.$$

Define $\overline{q}_{i,t} \triangleq \frac{\sum_{\tau=1}^{t} \mathbb{I}(i \in S_\tau, i_\tau = i)}{T_{i,t}} = \frac{\sum_{s=1}^{t} \mathbb{I}(i \in S_\tau) \cdot \mathbb{I}(i_\tau = i)}{\sum_{\tau=1}^{t} \mathbb{I}(i \in S_\tau)}$. Note that $\mathbb{I}(i_\tau = i)$ is a random variable that follows Bernoulli distribution $\text{Ber}(q_i)$, and one can only observe $\mathbb{I}(i_\tau = i)$ when $i \in S_\tau$, let $\overline{x}_{i,j}$ denote the mean value of first $j$ i.i.d. observations of $\mathbb{I}(i_s = i)$. Then, we have

$$\mathbb{P}\left(\left|\overline{q}_{i,t} - q_i\right| > \sqrt{\frac{\log T}{T_{i,t}}}\right) = \sum_{j=0}^{t} \mathbb{P}\left(\left|\overline{q}_{i,t} - q_i\right| > \sqrt{\frac{\log T}{T_{i,t}}}, \; T_{i,t} = j\right)$$

$$\le \sum_{j=0}^{t} \mathbb{P}\left(\left|\overline{x}_{i,j} - q_i\right| > \sqrt{\frac{\log T}{j}}\right)$$

$$\le \sum_{j=0}^{t} 2\exp(-2\log T)$$

$$\le \frac{2}{T}.$$

Where in the first inequality, the event $\left\{\left|\overline{q}_{i,t} - q_i\right| > \sqrt{\frac{\log T}{T_{i,t}}}, \; T_{i,t} = j\right\}$ indicates $\left\{\left|\overline{x}_{i,j} - q_i\right| > \sqrt{\frac{\log T}{j}}\right\}$, and the second inequality follows from Hoeffding's inequality.

We then bound the second term in (33)

$$\sum_{t=1}^{T} \mathbb{E}\left[\delta_{p_t} \mathbb{I}(A_t^c)\right] \le \sum_{t=1}^{T} \mathbb{E}\left[\mathbb{I}(A_t^c)\right]$$

$$\le \sum_{t=1}^{T} \sum_{i=1}^{m} \mathbb{P}\left(\left|\overline{q}_{i,t} - q_i\right| > \sqrt{\frac{\log T}{T_{i,t}}}\right)$$

$$\le \sum_{t=1}^{T} \sum_{i=1}^{m} \frac{2}{T}$$

$$\le 2m.$$

Define event $H_t \triangleq \left\{0 < \delta_{p_t} < 2\sum_{i \in S_t} \sqrt{\frac{\log T}{T_{i,t-1}}}\right\}$. By Lemma C.3, we know that

$$\mathbb{I}(A_{t-1}, \delta_{p_t} > 0) \implies \mathbb{I}\left(0 < \delta_{p_t} < \sum_{i \in S_t} 2\sqrt{\frac{\log T}{T_{i,t-1}}}\right) = \mathbb{I}(H_T).$$

It remains to prove the upper bound for $\sum_{t=1}^{T} \mathbb{E}\left[\delta_{p_t} \mathbb{I}(A_T)\right]$.

For $t \in \{1, \ldots, T\}$ and $k \in \mathbb{Z}_+$, let

$$m_{k,t} \triangleq \begin{cases} \alpha_k \left(\frac{m}{\delta_{p_t}}\right)^2 \log T, & \delta_{p_t} > 0, \\ +\infty, & \delta_{p_t} = 0, \end{cases}$$

and

$$A_{k,t} \triangleq \{i \in S_t : T_{i,t-1} \leq m_{k,t}\}.$$

Then, we define an event

$$\mathcal{G}_{k,t} \triangleq \{|A_{k,t}| \geq \beta_k m\},$$

which means "In the $t$-th round, at least $\beta_k m$ types in $S_t$ has been observed at most $m_{k,t}$ times".

Then, by Lemma C.5, we have

$$\sum_{t=1}^{T} \mathbb{I}(\mathcal{H}_t) \cdot \delta_{p_t} \leq \sum_{k=1}^{\infty} \sum_{t=1}^{T} \mathbb{I}\left(\mathcal{G}_{k,t}, \delta_{p_t} > 0\right) \cdot \delta_{p_t}.$$

For $i \in [m], k \in \mathbb{Z}_+, t \in [T]$, define an event

$$\mathcal{G}_{i,k,t} \triangleq \mathcal{G}_{k,t} \cap \{i \in S_t, T_{i,t-1} \leq m_{k,t}\}.$$

Then by the definitions of $\mathcal{G}_{k,t}$ and $\mathcal{G}_{i,k,t}$ we have

$$\mathbb{I}\left(\mathcal{G}_{k,t}, \delta_{p_t} > 0\right) \leq \frac{1}{\beta_k m} \sum_{i \in E_{\mathrm{B}}} \mathbb{I}\left(\mathcal{G}_{i,k,t}, \delta_{p_t} > 0\right).$$

Therefore,

$$\sum_{t=1}^{T} \mathbb{I}(\mathcal{H}_t) \cdot \delta_{p_t} \leq \sum_{i \in E_{\mathrm{B}}} \sum_{k=1}^{\infty} \sum_{t=1}^{T} \mathbb{I}\left(\mathcal{G}_{i,k,t}, \delta_{p_t} > 0\right) \cdot \frac{\delta_{p_t}}{\beta_k m}.$$

For any price function $p$, define $\delta_p \triangleq \mathrm{rev}(p^\star) - \mathrm{rev}(p)$. If $\delta_p > 0$, we call it a "bad" price. Let $E_B \triangleq \{i \in [m] : \text{type } i \text{ would make a purchase at least one bad price}\}$.

For each type $i \in E_{\mathrm{B}}$, suppose $i$ is contained in $N_i$ bad prices $p_{i,1}^{\mathrm{B}}, p_{i,2}^{\mathrm{B}}, \ldots, p_{i,N_i}^{\mathrm{B}}$. Let $\delta_{i,l} \triangleq \delta_{p_{i,l}^{\mathrm{B}}} \ (l \in [N_i])$. Without loss of generality, we assume $\delta_{i,1} \geq \delta_{i,2} \geq \cdots \geq \delta_{i,N_i}$. Let $\delta_{i,\min} \triangleq \delta_{i,N_i}$. For convenience, we also define $\delta_{i,0} = +\infty$, i.e., $\alpha_k \left(\frac{2m}{\delta_{i,0}}\right)^2 = 0$. Then, we have

$$\sum_{t=1}^{T} \mathbb{I}\left(\mathcal{H}_t\right) \delta_{p_t}$$

$$\leq \sum_{i \in E_{\mathrm{B}}} \sum_{k=1}^{\infty} \sum_{t=1}^{T} \mathbb{I}\left(\mathcal{G}_{i,k,t}, \delta_{p_t} > 0\right) \frac{\delta_{p_t}}{\beta_k m}$$

$$= \sum_{i \in E_{\mathrm{B}}} \sum_{k=1}^{\infty} \sum_{t=1}^{T} \sum_{l=1}^{N_i} \mathbb{I}\left(\mathcal{G}_{i,k,t}, p_t = p_{i,l}^{\mathrm{B}}\right) \frac{\delta_{p_t}}{\beta_k m}$$

$$= \sum_{i \in E_{\mathrm{B}}} \sum_{k=1}^{\infty} \sum_{t=1}^{T} \sum_{l=1}^{N_i} \mathbb{I}\left(\mathcal{G}_{i,k,t}, p_t = p_{i,l}^{\mathrm{B}}\right) \frac{\delta_{i,l}}{\beta_k m}$$

$$\leq \sum_{i \in E_{\mathrm{B}}} \sum_{k=1}^{\infty} \sum_{t=1}^{T} \sum_{l=1}^{N_i} \mathbb{I}\left(T_{i,t-1} \leq m_{k,t},\, p_t = p_{i,l}^{\mathrm{B}}\right) \frac{\delta_{i,l}}{\beta_k m}$$

$$= \sum_{i \in E_{\mathrm{B}}} \sum_{k=1}^{\infty} \sum_{t=1}^{T} \sum_{l=1}^{N_i} \mathbb{I}\left(T_{i,t-1} \leq \alpha_k \left(\frac{2m}{\delta_{i,l}}\right)^2 \log T,\, p_t = p_{i,l}^{\mathrm{B}}\right) \frac{\delta_{i,l}}{\beta_k m}$$

$$= \sum_{i \in E_{\mathrm{B}}} \sum_{k=1}^{\infty} \sum_{t=1}^{T} \sum_{l=1}^{N_i} \sum_{j=1}^{l} \mathbb{I}\left(\alpha_k \left(\frac{2m}{\delta_{i,j-1}}\right)^2 \log T < T_{i,t-1} \leq \alpha_k \left(\frac{2m}{\delta_{i,j}}\right)^2 \log T,\, p_t = p_{i,l}^{\mathrm{B}}\right) \frac{\delta_{i,l}}{\beta_k m}$$

$$\leq \sum_{i \in E_{\mathrm{B}}} \sum_{k=1}^{\infty} \sum_{t=1}^{T} \sum_{l=1}^{N_i} \sum_{j=1}^{l} \mathbb{I}\left(\alpha_k \left(\frac{2m}{\delta_{i,j-1}}\right)^2 \log T < T_{i,t-1} \leq \alpha_k \left(\frac{2m}{\delta_{i,j}}\right)^2 \log T,\, p_t = p_{i,l}^{\mathrm{B}}\right) \frac{\delta_{i,j}}{\beta_k m}$$

$$\leq \sum_{i \in E_{\mathrm{B}}} \sum_{k=1}^{\infty} \sum_{t=1}^{T} \sum_{l=1}^{N_i} \sum_{j=1}^{N_i} \mathbb{I}\left(\alpha_k \left(\frac{2m}{\delta_{i,j-1}}\right)^2 \log T < T_{i,t-1} \leq \alpha_k \left(\frac{2m}{\delta_{i,j}}\right)^2 \log T,\, p_t = p_{i,l}^{\mathrm{B}}\right) \frac{\delta_{i,j}}{\beta_k m}$$

$$\leq \sum_{i \in E_{\mathrm{B}}} \sum_{k=1}^{\infty} \sum_{t=1}^{T} \sum_{j=1}^{N_i} \mathbb{I}\left(\alpha_k \left(\frac{2m}{\delta_{i,j-1}}\right)^2 \log T < T_{i,t-1} \leq \alpha_k \left(\frac{2m}{\delta_{i,j}}\right)^2 \log T,\, i \in S_t\right) \frac{\delta_{i,j}}{\beta_k m}$$

$$\leq \sum_{i \in E_{\mathrm{B}}} \sum_{k=1}^{\infty} \sum_{j=1}^{N_i} \left(\alpha_k \left(\frac{2m}{\delta_{i,j}}\right)^2 \log T - \alpha_k \left(\frac{2m}{\delta_{i,j-1}}\right)^2 \log T\right) \frac{\delta_{i,j}}{\beta_k m}$$

$$= 4m \left(\sum_{k=1}^{\infty} \frac{\alpha_k}{\beta_k}\right) \log T \cdot \sum_{i \in E_{\mathrm{B}}} \sum_{j=1}^{N_i} \left(\frac{1}{\delta_{i,j}^2} - \frac{1}{\delta_{i,j-1}^2}\right) \delta_{i,j}$$

$$\leq 1068 m \log T \cdot \sum_{i \in E_{\mathrm{B}}} \sum_{j=1}^{N_i} \left(\frac{1}{\delta_{i,j}^2} - \frac{1}{\delta_{i,j-1}^2}\right) \delta_{i,j},$$

where the last inequality is due to Lemma C.4. Finally, for each $i \in E_{\mathrm{B}}$ we have

$$\sum_{j=1}^{N_i} \left(\frac{1}{\delta_{i,j}^2} - \frac{1}{\delta_{i,j-1}^2}\right) \delta_{i,j} = \frac{1}{\delta_{i,N_i}} + \sum_{j=1}^{N_i-1} \frac{1}{\delta_{i,j}^2} \left(\delta_{i,j} - \delta_{i,j+1}\right)$$

$$\leq \frac{1}{\delta_{i,N_i}} + \int_{\delta_{i,N_i}}^{\delta_{i,1}} \frac{1}{x^2} \, dx$$

$$= \frac{2}{\delta_{i,N_i}} - \frac{1}{\delta_{i,1}}$$

$$\leq \frac{2}{\delta_{i,\min}}.$$

It follows that

$$\sum_{t=1}^{T} \mathbb{I}(\mathcal{H}_t) \cdot \delta_{p_t} \leq 1068 m \log T \cdot \sum_{i \in E_{\mathrm{B}}} \frac{2}{\delta_{i,\min}} = m \sum_{i \in E_{\mathrm{B}}} \frac{2136}{\delta_{i,\min}} \log T \tag{34}$$

So far, the distribution-dependent regret bound is proven. To prove the distribution-independent bound, we decompose $\sum_{t=1}^{T} \mathbb{I}(\mathcal{H}_t) \cdot \delta_{p_t}$ into two parts:

$$\sum_{t=1}^{T} \mathbb{I}(\mathcal{H}_t) \cdot \delta_{p_t} = \sum_{t=1}^{T} \mathbb{I}\left(\mathcal{H}_t,\, \delta_{p_t} \leq \epsilon\right) \cdot \delta_{p_t} + \sum_{t=1}^{T} \mathbb{I}\left(\mathcal{H}_t,\, \delta_{p_t} > \epsilon\right) \cdot \delta_{p_t}$$

$$\leq \epsilon T + \sum_{t=1}^{T} \mathbb{I}\left(\mathcal{H}_t,\, \delta_{p_t} > \epsilon\right) \cdot \delta_{p_t},$$

where $\epsilon > 0$ is a constant to be determined. The second term can be bounded in the same way as in the proof of the distribution-dependent regret bound, except that we only consider the case

$\delta_{p_t} > \epsilon$. (For each type $i \in E_{\mathrm{B}}$, suppose $i$ is contained in $N_i$ bad prices $p_{i,1}^{\mathrm{B}}, p_{i,2}^{\mathrm{B}}, \ldots, p_{i,N_i}^{\mathrm{B}}$. Let $\delta_{i,l} \triangleq \delta_{p_{i,l}^{\mathrm{B}}}$ $(l \in [N_i])$ satisfies $\delta_{i,1} \geq \delta_{i,2} \geq \ldots \geq \delta_{i,N_i} \geq \epsilon$. Also let $\delta_{i,\min} \triangleq \delta_{i,N_i}$.) Thus, we can replace (34) by

$$\sum_{t=1}^{T} \mathbb{I}\left(\mathcal{H}_t, \delta_{p_t} > \epsilon\right) \cdot \delta_{p_t} \leq m \cdot \sum_{i \in E_{\mathrm{B}}, \delta_{i,\min} > \epsilon} \frac{2136}{\delta_{i,\min}} \log T \leq \frac{2136m^2}{\epsilon} \log T.$$

It follows that

$$\sum_{t=1}^{T} \mathbb{I}(\mathcal{H}_t) \cdot \delta_{S_t} \leq \epsilon T + \frac{2136m^2}{\epsilon} \log T.$$

Finally, letting $\epsilon = \sqrt{\frac{2136m^2 \log T}{T}}$, we get

$$\sum_{t=1}^{T} \mathbb{I}(\mathcal{H}_t) \cdot \delta_{S_t} \leq 2\sqrt{2136m^2 T \log T} \leq 93\sqrt{m^2 T \log T}.$$

$\square$

**Lemma C.2.** *Under good event $A_t$, for any price function $p$, let $S_p$ denote the set of types who would purchase at price $p$, then we have*

$$\forall t \in [T], \quad \mathrm{rev}(p) \leq \widehat{\mathrm{rev}}_t(p) \leq \mathrm{rev}(p) + \sum_{i \in S_p} 2\sqrt{\frac{\log T}{T_{i,t}}}.$$

*Proof of Lemma C.2.* When $A_t$ happens,

$$q_i \leq \widehat{q}_{i,t} \leq q_i + 2\sqrt{\frac{\log T}{T_{i,t}}},$$

for all $i \in [m]$.

Therefore, we have

$$\widehat{\mathrm{rev}}_t(p) = \sum_{i=1}^{m} \widehat{q}_{i,t} \cdot r(i,p) \geq \sum_{i=1}^{m} q_i \cdot r(i,p) = \mathrm{rev}(p)$$

and

$$\widehat{\mathrm{rev}}_t(p) = \sum_{i=1}^{m} \widehat{q}_{i,t} \cdot r(i,p) \leq \sum_{i=1}^{m} \left(q_i + 2\sqrt{\frac{\log T}{T_{i,t}}}\right) \cdot r(i,p) \leq \mathrm{rev}(p) + \sum_{i \in S_p} 2\sqrt{\frac{\log T}{T_{i,t}}}.$$

The last inequality is by $r(i,p) \leq 1$. $\square$

**Lemma C.3.** *For each $t \in [T]$, under good event $A_{t-1}$, the following inequality holds,*

$$\delta_{p_t} \triangleq \mathrm{rev}(p^\star) - \mathrm{rev}(p_t) \leq 2\sum_{i \in S_t} \sqrt{\frac{\log T}{T_{i,t-1}}}.$$

*Proof of Lemma C.3.* When $A_{t-1}$ happens, by Lemma C.2,

$$\mathrm{rev}(p^\star) \leq \widehat{\mathrm{rev}}_{t-1}(p^\star),$$

$$\mathrm{rev}(p_t) \geq \widehat{\mathrm{rev}}_{t-1}(p_t) - 2\sum_{i \in S_t} \sqrt{\frac{\log T}{T_{i,t-1}}}.$$

It then follows that,

$$\delta_{p_t} = \text{rev}(p^\star) - \text{rev}(p_t) \leq \widehat{\text{rev}}_{t-1}(p^\star) - \left( \widehat{\text{rev}}_{t-1}(p_t) - 2 \sum_{i \in S_t} \sqrt{\frac{\log T}{T_{i,t-1}}} \right)$$

Since $p_t = \operatorname{argmax}_{p \in \overline{\mathcal{P}}} \widehat{\text{rev}}_{t-1}(p)$, we have

$$\widehat{\text{rev}}_{t-1}(p_t) \geq \widehat{\text{rev}}_{t-1}(p^\star).$$

$\square$

**Lemma C.4** (Theorem 4 of Kveton et al. [37])**.** *We can choose $\{\alpha_k\}_{k \geq 0}$ and $\{\beta_k\}_{k \geq 0}$, which satisfy the following properties: $\{\alpha_k\}_{k \geq 0}$ and $\{\beta_k\}_{k \geq 0}$ are positive and*

$$\alpha_1 > \alpha_2 > \dots \quad \text{and} \quad 1 = \beta_0 > \beta_1 > \beta_2 > \dots,$$

*such that $\lim_{k \to \infty} \alpha_k = \lim_{k \to \infty} \beta_k = 0$. Moreover,*

$$\sqrt{6} \sum_{k=1}^{\infty} \frac{\beta_{k-1} - \beta_k}{\sqrt{\alpha_k}} \leq 1, \quad \text{and} \quad \sum_{k=1}^{\infty} \frac{\alpha_k}{\beta_k} < 267.$$

**Lemma C.5.** *On round $t$, if event $\mathcal{H}_t$ happens, then at least one event $\mathcal{G}_{k,t}$, $k \in \mathbb{Z}_+$ happens, where*

$$\mathcal{G}_{k,t} \triangleq \{|A_{k,t}| \geq \beta_k m\}, \quad \text{where } A_{k,t} \triangleq \{i \in S_t : T_{i,t-1} \leq m_{k,t}\},$$

*and $m_{k,t} = \alpha_k \left( \frac{m}{\delta_{p_t}} \right)^2 \log T$ when $\delta_{p_t} > 0$ and $+\infty$ otherwise.*

*Proof of Lemma C.5.* Assume that $\mathcal{H}_t$ happens and that none of $\mathcal{G}_{1,t}, \mathcal{G}_{2,t}, \dots$ happens. Then $|A_{k,t}| < \beta_k m$ for all $k \in \mathbb{Z}_+$. Let $A_{0,t} = S_t$ and $\bar{A}_{k,t} = S_t \backslash A_{k,t}$ for $k \in \mathbb{Z}_+ \cup \{0\}$. Thus $\bar{A}_{k-1,t} \subseteq \bar{A}_{k,t}$ for all $k \in \mathbb{Z}_+$. Note that $\lim_{k \to \infty} m_{k,t} = 0$. Thus there exists $N \in \mathbb{Z}_+$ such that $\bar{A}_{k,t} = S_t$ for all $k \geq N$, and then we have $S_t = \bigcup_{k=1}^{\infty} \left( \bar{A}_{k,t} \backslash \bar{A}_{k-1,t} \right)$. Finally, note that for all $i \in \bar{A}_{k,t}$, we have $T_{i,t-1} > m_{k,t}$. Therefore

$$
\begin{aligned}
\sum_{i \in S_t} \frac{1}{\sqrt{T_{i,t-1}}} &= \sum_{k=1}^{\infty} \sum_{i \in \bar{A}_{k,t} \backslash \bar{A}_{k-1,t}} \frac{1}{\sqrt{T_{i,t-1}}} \leq \sum_{k=1}^{\infty} \sum_{i \in \bar{A}_{k,t} \backslash \bar{A}_{k-1,t}} \frac{1}{\sqrt{m_{k,t}}} \\
&= \sum_{k=1}^{\infty} \frac{\left| \bar{A}_{k,t} \backslash \bar{A}_{k-1,t} \right|}{\sqrt{m_{k,t}}} = \sum_{k=1}^{\infty} \frac{|A_{k-1,t} \backslash A_{k,t}|}{\sqrt{m_{k,t}}} = \sum_{k=1}^{\infty} \frac{|A_{k-1,t}| - |A_{k,t}|}{\sqrt{m_{k,t}}} \\
&= \frac{|S_t|}{\sqrt{m_{1,t}}} + \sum_{k=1}^{\infty} |A_{k,t}| \left( \frac{1}{\sqrt{m_{k+1,t}}} - \frac{1}{\sqrt{m_{k,t}}} \right) \\
&< \frac{m}{\sqrt{m_{1,t}}} + \sum_{k=1}^{\infty} \beta_k m \left( \frac{1}{\sqrt{m_{k+1,t}}} - \frac{1}{\sqrt{m_{k,t}}} \right) \\
&= \sum_{k=1}^{\infty} \frac{(\beta_{k-1} - \beta_k) m}{\sqrt{m_{k,t}}}.
\end{aligned}
$$

Under event $\mathcal{H}_t$, we have

$$
\begin{aligned}
\delta_{p_t} &\leq \sum_{i \in S_t} 2 \sqrt{\frac{\log T}{T_{i,t-1}}} = 2\sqrt{\log T} \cdot \sum_{i \in S_t} \frac{1}{\sqrt{T_{i,t-1}}} \\
&< 2\sqrt{\log T} \cdot \sum_{k=1}^{\infty} \frac{(\beta_{k-1} - \beta_k) m}{\sqrt{m_{k,t}}} = 2 \sum_{k=1}^{\infty} \frac{\beta_{k-1} - \beta_k}{\sqrt{\alpha_k}} \cdot \delta_{p_t} \leq \delta_{p_t},
\end{aligned}
$$

where the last inequality is due to Lemma C.4. We reach a contradiction here, hence the lemma follows. $\square$

# D  Miscellaneous

## D.1  Notations

The following table contains the notations used in this paper.

| Notation | Meaning |
|---|---|
| $N$ | The total amount of data. |
| $n \in [N]$ | The number of data. |
| $m$ | The number of types. |
| $p : [N] \to [0,1]$ | A price curve. |
| $\overline{\mathcal{P}}$ | A set of discretized price curves. |
| $v_i : [N] \to [0,1]$ | The valuation curve for type $i \in [m]$. |
| $\mathcal{V} = \{v_i : i \in [m]\}$ | The set of all valuation curves. |
| $n_{i,p}$ | The amount of data type $i \in [m]$ purchases at price curve $p$. |
| $r(i,p) = p(n_{i,p})$ | The revenue from type $i \in [m]$ under price curve $p$. |
| $q = (q_1, q_2, \ldots, q_m)$ | The type distribution. |
| $\mathrm{rev}(p)$ | The expected revenue under price $p$. |
| $i_t \in [m]$ | The type of buyer on round $t \in [T]$. |
| $p_t : [N] \to [0,1]$ | The price curve on round $t \in [T]$. |
| $S_t$ | The set of types that would make a purchase at price $p_t$. |
| $S_p$ | The set of types that would make a purchase at price $p$. |
| $T_{i,t} \triangleq \sum_{\tau=1}^{t} \mathbb{I}(i \in S_\tau)$ | The number of times that type $i$ appears in set $S_\tau$ for $\tau \in \{1, \ldots, t\}$. |
| $\mathcal{P} = \{p \in [N] \to [0,1] : p(0) = 0\}$ | The set of all pricing curves. |
| $L$ | Smoothness constant of valuation curves. |
| $J$ | Diminishing return constant of valuation curves. |

Table 3: Table of notations.

