# OpenReview forum: "Learning to Price Homogeneous Data"
_NeurIPS.cc/2024/Conference — NeurIPS 2024 poster_

### Official Review · Reviewer_oJtn · 2024-07-08

**Soundness:** 3
**Presentation:** 3
**Contribution:** 3
**Rating:** 6
**Confidence:** 3

**Summary:**

The paper studies an online learning problem for pricing homogeneous data, where the seller needs to offer a price function/curve based on the data size and learn the arrival probabilities of different customer types to maximize cumulative revenue. The paper first analyzes the structure of the optimal price function when there is a finite number of customer types with known arrival probabilities. It introduces new discretization schemes to achieve better dependence on the approximation error compared to existing methods. When the arrival probabilities are unknown (i.e., in the online learning setting), the paper develops algorithms for both stochastic and adversarial settings with theoretically bounded regret.

**Strengths:**

The paper is well-written and introduces an interesting online learning problem in data pricing. It explores the unique structure of the optimal pricing curve in this problem and provides algorithms to solve this task under both stochastic and adversarial settings with theoretical regret bounds. The paper also discusses its technical contributions in deriving theoretical results.

**Weaknesses:**

(1) Computational Complexity: The provided algorithms (Algorithm 3 and 4) have computational complexity depending on the horizon $T$, which can be large in practice.

(2) Lack of Numerical Validations: Since the problem is motivated by a practical pricing scenario, numerical experiments could help validate the efficiency of the algorithms, especially given the potentially large computational complexity.

**Questions:**

1. In Theorems 4.1 and 5.1, can  the discretization additive approximation $O(\frac{1}{\sqrt{T}})$ be replaced by $O(\frac{1}{\sqrt{t}})$ while maintaining the same order of regret (i.e., the discretization level changes across the periods)? The intuition is that when the estimations of $q$'s  are inaccurate, we may only need an inaccurate discretization of the price curve, potentially reducing computational complexity.
2. How can the Upper Confidence Bound (UCB) on $q$'s facilitate exploration in learning $q$'s (i.e., more explorations)? My understanding is that to learn $q$'s (i.e., collecting samples of $q$'s), we need to encourage purchases by setting a price curve lower than the estimated optimal one. However, the UCB on $q$'s does not obviously encourage lower prices.

**Limitations:**

See weaknesses.

---

> ### Author Rebuttal · Authors · 2024-08-07
>
> ### ***Weakness 1. Computational complexity depending on the horizon $T$.***
> As mentioned in line 58, achieving sublinear regret in online learning requires choosing $\epsilon$ that vanishes with longer time horizons, i.e., $\epsilon \to 0$ as $T \to \infty$. In Theorems 4.1 and 5.1, we choose $\epsilon = 1/\sqrt{T}$ to achieve regret upper bounds $\tilde{O}(m\sqrt{T})$ and $\tilde{O}(m^{3/2}\sqrt{T})$, respectively. However, if one does not want the per-round complexity to scale with $T$, one can fix the error as a constant $\epsilon$. Then the discretization size $|\overline{\mathcal{P}}|$ would be $\left(\frac{N}{\epsilon}\right)^m$, which is no longer dependent on $T$. With a slight sacrifice of the seller's revenue, the regret bound in Theorem 4.1 would be $2\epsilon T + 93m\sqrt{T \log T}$, and the regret bound in Theorem 5.1 would be $2\epsilon T + 3m\sqrt{T \log |\overline{\mathcal{P}}|}$.
>
> This is standard in continuous bandit, we choose the discretization size to scale with $T$ to achieve the lowest possible regret bound, which causes large computation complexity at the same time (for a classic example, see [1]).
>
> ### ***Weakness 2. Lack of Numerical Validations.***
> See the common response above.
>
> ### ***Question 1. Can we reduce discretization additive approximation?***
> Yes, it is possible to replace the size of the discretization from $O\left(1/\sqrt{T}\right)$ with $O\left(1/\sqrt{t} \right)$ in the stochastic setting. This will reduce the computational complexity, but only by a constant factor, so it does not give a fundamentally different result. See the common response as to why we think large computational complexity is unavoidable for this setting.
>
> In the adversarial setting, we believe that it is not possible to reduce $O\left(1/\sqrt{T}\right)$ to $O\left(1/\sqrt{t}\right)$.
>
> ### ***Question 2. How can the Upper Confidence Bound (UCB) on type distribution $q$ facilitate exploration in learning $q$?***
>
> If type $i$ has not been explored enough before, then $T_{i,t}$ is small, and the upper confidence bound for $q_i$ is large. We let $S_p$ denote the set of types that would make a purchase under price $p$. Then for all prices $p$ such that $i\in S_p$, their UCBs for revenue (defined in Eq. 7) are large. Therefore, the algorithm tends to choose $p$ satisfying $i\in S_p$ in the following rounds, leading to an increase in $T_{i,t}$. This encourages exploration of type $i$.
>
> ### ***Reference***
>
> [1] R. Kleinberg and T. Leighton. The value of knowing a demand curve: Bounds on regret for online
> posted-price auctions. FOCS 2003.

---

> > ### Comment · Reviewer_oJtn · 2024-08-11
> >
> > Thank you for your response. It addresses my concerns and I have raised the score.

---

> > > ### Author Response · Authors · 2024-08-11
> > >
> > > Thank you again for your comments. Based on your comments, we will clarify our contributions and techniques in future revisions.

---

> ### Author Response · Authors · 2024-08-11
> **Sincerely looking forward to your reply**
>
> Thank you again for your feedback. With the discussion period ending in two days, we would appreciate knowing if our response has adequately addressed your key questions and concerns.

---

### Official Review · Reviewer_3dYq · 2024-07-13

**Soundness:** 3
**Presentation:** 3
**Contribution:** 2
**Rating:** 5
**Confidence:** 4

**Summary:**

Motivated by the emergence of data marketplaces, this paper studies an online data pricing problem involving $ N $ homogeneous data points and $ m $ types of buyers in the market. Specifically, it assumes that each type of buyer has a specific value function $ v_i: [N] \rightarrow [0, 1] $. While the sellers know all the value curves $ v_i $, they do not know the distribution of buyers. The sellers need to choose a pricing curve $ p \in \mathcal{P}: [N] \rightarrow [0, 1] $ at each time period. To address this online pricing problem, the authors develop novel discretization schemes to approximate any pricing curve. To minimize the discretization size, they propose assumptions such as smoothness and diminishing returns. To solve the online learning problem, they build on classical algorithms like UCB (Upper Confidence Bound) and FTPL (Follow-The-Perturbed-Leader), providing corresponding regret bounds. These advancements contribute to more effective and efficient online data pricing strategies.

**Strengths:**

1. The topic related to marketplaces is very interesting. This paper is well-written and easy to follow.

2. The authors propose three types of price discretization schemes under different assumptions: monotonic valuations, smooth monotonic valuations, and monotonic valuations under diminishing returns. These schemes, based on Algorithm 1, significantly reduce the discretization size. While Algorithm 1 is an existing technique, the proposed price discretization schemes are non-trivial and add substantial value.

3. The proof provided is rigorous, and the theoretical results demonstrate sublinear regret with respect to $T$.

**Weaknesses:**

1. I think Sections 3 and 4-5 are independent. Section 3 primarily introduces the price discretization schemes, while Sections 4-5 discuss online learning algorithms for both settings. In my opinion, the main contribution of your paper lies in Section 3, as the techniques in Sections 4-5 seem to be standard.

2. You claim that your analysis when constructing the UCB in this way is non-trivial since the types are observed only if they make a purchase. However, this seems common. Could you elaborate on the additional difficulties for the UCB algorithms in your case? Additionally, in line 266, maintaining UCBs for the type distribution appears to be a standard approach, and I don't see this as a significant contribution.

3. Section 5 is a little confusing. Why is it necessary to add perturbation?  Maybe you should give me some intuition. Moreover, in line 320, why is \( r_t(p) \) an upper bound on \( p(n_{i_t,p}) \)?

**Questions:**

1. While I understand that Algorithm 1 is for price discretization, and the valuation space is discretized, it remains unclear how the price set \( \bar{\mathcal{P}} \) is determined. Could you provide more details on this process?
2. Although you have provided the regret bound, additional discussion on its tightness would be valuable. Can you elaborate on the tightness of the regret bound and compare it with existing results?
3. Sections 4-5 discuss the online learning algorithms. Could you highlight the specific difficulties you encountered in these sections? What new techniques did you develop? What are your unique contributions? Clarifying these points will help distinguish your work from standard techniques.

**Limitations:**

As mentioned above, the studied topic is very interesting. However, I am eager to see more detailed statements regarding the unique contributions, particularly in Sections 4-5.

---

> ### Author Rebuttal · Authors · 2024-08-07
>
> ### ***Weakness 1. Main contribution of our paper lies in Section 3, as the techniques in Sections 4-5 seem to be standard.***
> See the common response above, Novelty in Sections 4, 5.
>
> ### ***Weakness 2. Contribution in Section 4.***
> See the common response above.
>
> ### ***Weakness 3. Why is it necessary to add perturbation.***
> Adding perturbation is standard in the FTPL method as it ensures robustness against adversaries and helps bound the regret. For a classic introduction to FTPL, please refer to the following paper:
>
> - Kalai *et. al.*, Efficient algorithms for online decision problems. Journal of Computer and System Sciences, 2005.
>
> However, for reasons explained in the common response, the vanilla FTPL does not work under asymmetric feedback. Especially, in line 320, $r_t(p)\geq p(n_{i_t,p})$ is by definition (see lines 6-7 of Algorithm 4). When the buyer makes a purchase, $r_t(p)=p(n_{i_t,p})$; otherwise, since $i_t\in S_t^c$, we have $r_t(p)=\sum_{i\in S_t^c}p(n_{i,p})\geq p(n_{i_t,p})$. For specific difficulties and new techniques, please refer to our common response above.
>
> ### ***Question 1. Construction of the discretization set.***
> First, in Lemma 3.1, we prove that for any non-decreasing price curve $p $, there exists an $m $-step price that yields at least the same revenue as $ p $. Here, an $m$-step function is a non-decreasing function where $ p(n+1) $ and $p(n) $ differ at most $m $ times, i.e., at most $m $ jumps. See lines 216-223.
>
> Then in Algorithm 1, let $W$ be the discretization of the valuation space $[0,1]$. We define the discretization set $ \overline{\mathcal{P}} $ to be the class of all $m $-step functions mapping $[N]$ to $ W $. This means that we select all possible $m $-step functions that have a domain in $[N]$ and take values in $ W $.
>
> ### ***Question 2. Tightness of the regret bound and compare it with existing results.***
> See the common response above.
>
> ### ***Question 3. Unique contributions in Sections 4 and 5.***
> See the common response above.

---

> ### Author Response · Authors · 2024-08-11
> **Sincerely looking forward to your reply**
>
> Thank you again for your feedback. With the discussion period ending in two days, we would appreciate knowing if our response has adequately addressed your key questions and concerns.

---

### Official Review · Reviewer_kUrw · 2024-07-16

**Soundness:** 3
**Presentation:** 3
**Contribution:** 2
**Rating:** 7
**Confidence:** 4

**Summary:**

This paper considers a data pricing problem in which a seller has $N$ homogeneous data points they wish to sell access to.  The seller sets a price curve $p(n)$ which specifies the price a buyer must pay for access to $n$ data points for each $n \in [N]$.  Upon arrival, a buyer sees the price curve, and chooses an amount $n$ by maximizing their utility $u(n) = v(n) - p(n)$, where $v(n)$ is their valuation curve.  The buyer leaves without making a purchase if their utility is negative for all $n$.  It is assumed that there are $m$ buyer types, where all buyers of type $i$ have the same valuation curve $v_i$, and that there is a distribution $q$ over buyer types in which the arriving type is sampled i.i.d. from $q$ in each step.  The objective is then to find a price curve which maximizes the expected revenue over this distribution.  This paper focuses on online learning settings in which the distribution is unknown and must be learned or the arrival sequence is arbitrary and we want to compare to the best fixed price curve in hindsight, where now the goal is to minimize the regret.

Due to the nature of this problem (a relation to revenue maximization with unit-demand buyers), computing the exact optimal is intractable, so approximations are considered.  This is done by discretizing the space of valuation curves.  The main result of this paper are new discretization schemes for this problem which can accommodate various further assumptions such as smoothness or diminishing returns in the valuation curves.  These are then applied to develop algorithms with regret $\tilde{O}(m \sqrt{T})$ and $\tilde{O}(m^{3/2} \sqrt{T})$ in the stochastic and adversarial settings, respectively.

-----------------------

Edit: following the rebuttal some of my questions have been answered and my score has been raised.

**Strengths:**

Overall, this is a well-written paper which introduces a very interesting and relevant problem and gives a clean description of the proposed solutions.  While the techniques for online learning are based on fairly standard ideas (UCB and FTPL), they still require some small tricks that are particular to the new setting in this paper.

**Weaknesses:**

There are a few concerns:
 - Lack of tightness - it is not clear to what extent these results are tight as no lower bounds are given.
 - The finite type assumption - assuming both that the number of types is small (so that the complexity bounds are reasonable) and that all the types are known up front is somewhat limiting.  Anything to address either of these comments is a significant improvement.
 - Once given the discretization scheme, the online learning methods utilize fairly standard techniques.  It would be interesting if there are other approaches to this problem.
 - A lack of an experimental evaluation which might help to provide further evidence of the practicality of the proposed methods.

As a minor comment, the organization in the appendix is poor as it does not follow the order in the main paper (appendix C gives proofs for section 5, while appendix D gives proofs for section 4).  This leads to a confusion about notation, since the definition that $r(i, p ) = p(n_{i,p})$ isn't introduced until appendix D, but is used in appendix C.

**Questions:**

- Can anything be done to remove the assumption that the types are known up front?
 - Can any tight lower bounds be given for the stochastic and adversarial settings (i.e., in terms of both $m$ and T$)?

Minor comments:
 - In Eq. (22) in the appendix, it has $f_t(p^*)$ instead of $r_t(p^*)$.

**Limitations:**

The authors have made assumptions clear.

---

> ### Author Rebuttal · Authors · 2024-08-07
>
> ### ***Weakness 1, Question 2. Lower bounds and lack of tightness.***
> See the common response above.
>
> ### ***Weakness 2, Question 1. The finite type assumption. Can we remove the assumption that the types are known up front?***
> This is an interesting question. We did attempt to solve this problem, but it appears to be quite challenging, and we believe it is best left for future work. The setting we study here is novel and nontrivial, even if it is less general than assuming finite types and their valuation curves are unknown. Moreover, as we have explained in lines 133-138 (Example 1) and 164-168, it is also motivated by some practical use cases.
>
> ### ***Weakness 3. Once given the discretization scheme, our method uses fairly standard techniques.***
> See the common response above, Novelty in Sections 4, 5.
>
> ### ***Weakness 4. A lack of an experimental evaluation.***
> See the common response above.
>
> ### ***Organization of the appendix.***
> Yes, we agree that the appendix could be organized better and will reorganize it in the revision.

---

> > ### Comment · Reviewer_kUrw · 2024-08-07
> >
> > Thank you for the response here and the discussion above.  I agree that there is more subtlety to the techniques in this paper than I originally let on.  I have raised my score.

---

> ### Author Response · Authors · 2024-08-08
>
> Thank you so much! We will make the contributions clearer in the revision.

---

### Official Review · Reviewer_oHSK · 2024-07-22

**Soundness:** 3
**Presentation:** 2
**Contribution:** 3
**Rating:** 6
**Confidence:** 2

**Summary:**

This paper addresses the problem of exploration in pricing strategy. It proposes an algorithm and proves that it has lower regret compared to previous algorithms.

**Strengths:**

The setting is interesting, and the results in this paper seem to show improvement over previous algorithms.

**Weaknesses:**

1. One of the contributions of this paper is a better discretization scheme. An introduction to the insights that allowed you to design this improved discretization method would make the contribution clearer.

2. The construction of the confidence interval has room for improvement. The UCB in this paper is |q_i-q_{i,t}| \leq |log T/T_{i,t}|. Such an interval is typewise. I believe that using a joint confidence interval (for example, \sum T_{i,t} |q_i-q_{i,t}| \leq log T or similar form )or a similar form) could improve the dependency of the regret on $m$.

3. It is unclear whether Line 6 of Algorithm 3 is computationally efficient.

**Questions:**

See the 'Weakness' part

**Limitations:**

See the 'Weakness' part

---

> ### Author Rebuttal · Authors · 2024-08-07
>
> ### ***Weakness 1. Insights for discretization scheme.***
> First note that in order to get a discretization that approximates any price curve within $\epsilon$, the size of such a discretization should be $\tilde{O}\left( 2^N\epsilon^{-N} \right)$, which is clearly very large.
> Our first insight is that when there are only $m$ types and the valuations are monotone, we can reduce this to
> $m$ step functions (see Line 216-223), which reduces discretization size to $\tilde{O}\left(N^m\epsilon^{-m}\right)$.
> While this is better, it can still be quite bad.
> Hence, we explore two other assumptions satisfied by data.
> First, under smoothness, we are able to reduce the discretization size to
> $\tilde{O}\left(L^m \epsilon^{-2m}\right)$ by discretizing the data space $[0,N]$ uniformly ($L$ is the smoothness constant, see line 158).
> Second, under diminishing returns, we are able to reduce this to $\tilde{O}\left(J^m \epsilon^{-3m}\log^m N\right)$ ($J$ is the diminishing return constant, see line 162-163); in this case, we design a non-uniform discretization method to the data space $[0,N]$. The discretization needs to be denser near 0 to account for the diminishing returns structure.
>
> ### ***Weakness 2. Construction of confidence interval.***
> Here is the reason why we need a confidence interval for each of the $q_i$, where $q_i$ denotes the probability of type $i$. We first construct a confidence interval for type distribution $q = (q_1,\dots,q_m)$, then translate them to UCBs for the revenue (see line 760, Lemma D.2). Subsequently, these UCBs for the revenue are used to bound the regret (see equation 32 and Lemma D.3 in line 768).
>
> However, we are not aware of any union bounds which can improve the dependence on $m$. Could you please give us a concrete example of a union bound? That will help us give a better answer.
>
> ### ***Weakness 3. Computational complexity.***
> See the common response above.

---

> ### Author Response · Authors · 2024-08-11
> **Sincerely looking forward to your reply**
>
> Thank you again for your feedback. With the discussion period ending in two days, we would appreciate knowing if our response has adequately addressed your key questions and concerns.

---

> > ### Comment · Reviewer_oHSK · 2024-08-12
> > **Official Comment by Reviewer oHSK**
> >
> > Thank you for the response here and the discussion above. I have raised my score.

---

> > > ### Author Response · Authors · 2024-08-13
> > >
> > > Thank you again for your comments. Based on your comments, we will clarify our contributions and techniques in future revisions.

---

### Author Rebuttal · Authors · 2024-08-07

We thank all reviewers for their feedback. First, we would like to address common concerns raised by reviewers.

### ***On technical novelties***
While all reviewers agree on the novelty of our discretization scheme, there is a perception that Sections 4 and 5 simply apply UCB/FTPL to this discretization. However, there are nontrivial adaptations in the algorithm and the analysis to account for the asymmetric nature of the feedback. Specifically, the type is revealed only if a purchase is made, and feedback under one price curve can be shared across all prices.

- **Stochastic setting (Sec. 4)** If we naively apply UCB by only accounting for feedback for the chosen price curve, we get a $\sqrt{|\overline{\mathcal{P}}|T\log T}$ upper bound, where $|\overline{\mathcal{P}}|$ is the size of the price set, leading to poor, often exponential dependence on the number of types $m$. This is the bound if we only observe the reward for the prices that are actually pulled, but do not observe the types after purchase. Therefore, naively applying UCB is like bandit feedback.
 On the other extreme, had we been in an alternative setting where we observe the type regardless of purchase, this is like a full information feedback because once observe the type, we know the revenue for all prices. In this full information setting, UCB gives us $\sqrt{T\log|\overline{\mathcal{P}}|\log T}$ $\in \tilde{O}(\sqrt{mT})$ upper bound. We are in an intermediate regime between bandit feedback and full information: if a type purchases at one price curve, we know what they would have purchased at other price curves. However, if there is no purchase, we do not know the type of the buyer.

    We account for this asymmetric nature by noting that the key unknown is the type distribution. In the full information setting, we use the sample average of each type to estimate the type distribution and then translate this to confidence intervals on the revenue. In our setting, as the type is unknown if there was no purchase, we only count the types in $S_t$ if a particular type would have made a purchase anyway (Eq. 5, 6). The key challenge is in showing that the confidence intervals we have designed are valid, which requires a delicate analysis.

- **Adversarial setting (Sec. 5)** In the adversarial setting, we adapt the FTPL algorithm, which is for full information and does not apply to our setting directly. When there is a purchase, we set the reward similarly, but when there is no purchase, we do not observe $i_t$. Our approach in line 7 of Algorithm 4 is to set $r_t(p)=\sum_{i\in S_t^c}p(n_{i,p})$ for $p\in\overline{\mathcal{P}}$, where $S_t$ contains types who would have purchased in round $t$ had they appeared in that round.

    When the buyer does not purchase on round $t$, the seller knows that $i_t\in S_t^c$. We define the reward for each price function as $r_t(p)=\sum_{i\in S_t^c}p(n_{i,p})$, an upper bound on the true revenue $p(n_{i_t,p})$. For prices possessing the same set $S_p$ with $S_t$, this upper bound is tight, i.e., $r_t(p)=p(n_{i_t,p})=0$. As mentioned in lines 321-324, $r_t(p)$ deals with the uncertainty of not knowing the type on round $t$ by providing a large reward to prices that could have resulted in a purchase, encouraging exploration of such prices in future rounds.

### ***Lower bound***
Indeed, we were able to prove a $\Omega(\sqrt{T})$ lower bound in our setting. This is tight in $T$ but not in $m$, but we decided not to include it because one does not expect to do better than $\tilde{O}(\sqrt{T})$ anyway. (In hindsight, we think it might be better to include this, as the proof required some slightly different techniques to standard hypothesis testing arguments)

To get a tight dependence on $m$, we need to account for the asymmetric feedback in the proof of the lower bound. This appears to be challenging, and we are unaware of techniques to handle such feedback. The closest we could find is [2], who provide a $\sqrt{T\log K}$ lower bound for a $K$ armed bandit setting with one-sided feedback, but their techniques are not applicable here.

### ***Computational complexity***
Yes, our algorithm is computationally expensive (i.e., the running time depends exponentially on the number of types $m$), but this is inevitable as even the offline version of our problem is strongly NP-hard (see [1] and lines 449-463 in our paper). Our goal was to develop an algorithm that is efficient in the number of data points $N$ given that the number of types $m$ is a fixed constant, which is relevant in practical scenarios.

An interesting question is developing an online PTAS (Polynomial Time Approximation Scheme), an algorithm that guarantees sublinear regret with respect to an approximately optimal price with a fixed approximation factor. While this was not a focus of our paper, the offline PTAS from [1] can be generalized into our online setting using our online algorithms in Sections 4 and 5. In future revisions, we will address the issue of computational complexity more explicitly as shown in this rebuttal.

### ***Numerical evaluation***
While we agree that empirical evaluations would be helpful, we believe our theoretical contributions are valuable on their own. Furthermore, it is common and acceptable for theoretical papers at NeurIPS to not have empirical evaluations.

### ***Reference***

[1] Chawla et al., Pricing ordered items. STOC 2022.

[2] Zhao et al., Stochastic One-Sided Full-Information Bandit. ECML PKDD 2019.

---

### Decision · Program_Chairs · 2024-09-25

**Decision:**

Accept (poster)

**Comment:**

This paper on learning prices received positive grades from all the reviewers, yet none of them were that thrilled about the paper.

I went through it myself, and I must concur with them. This is a good and interesting paper (even though it's not mind-blowing, but not all papers have to be, thankfully).

So we decided to recommend acceptance.